# The heterogeneity of dermal mesenchymal cells reproduced in skin equivalents regulates barrier function and elasticity

Shun Kimura [ID][1,✉], Sachiko Sekiya[2,3], Sawa Yamashiro [ID][1], Tetsutaro Kikuchi [ID][2], Masatoshi Haga [ID][1] & Tatsuya Shimizu [ID][2]

## Abstract

The heterogeneity of dermal mesenchymal cells, including perivascular mesenchymal cells and papillary and reticular fibroblasts, plays critical roles in skin homeostasis. Herein, we present human skin equivalents (HSEs), in which pericytes, papillary fibroblasts, and reticular fibroblasts are spatially organized through autonomous three-cell interactions among epidermal keratinocytes, dermal fibroblasts, and vascular endothelial cells. The replication of dermal mesenchymal cell heterogeneity enhances skin functions, including epithelialization, epidermal barrier formation, and dermal elasticity, enabling in vitro evaluation of drug efficacy. Furthermore, ascorbic acid-induced epidermal turnover and synthesis of well-aligned extracellular matrix via perivascular niche cells play crucial roles in improving skin barrier function and elasticity. Therefore, HSEs with heterogeneous dermal mesenchymal cells may improve our understanding of the mechanisms underlying skin homeostasis through cell-to-cell communication and serve as a model to animal experiments for developing precision medicine.

**Keywords** Human Skin Equivalent; Fibroblasts Heterogeneity; Pericyte; Ascorbic Acid; Aging
**Subject Categories** Methods & Resources; Skin

## Introduction

Intercellular crosstalk based on cellular heterogeneity plays a crucial role in maintaining the morphological and functional homeostasis of organs and has garnered attention as a potential target for studies on precision medicine (Cha and Lee, 2020). Given that the integumentary organ system (IOS) is the most readily accessible organ system in the human body, the analysis of heterogeneity in human skin cells has established paradigms for scientific research in areas such as cell adhesion, inflammation, and tissue stem cells (Watt, 2014). The IOS, which includes skin, skin appendages, nerves, and blood vessels, plays essential roles in waterproofing, cushioning, protecting relatively deep tissues, excreting waste and thermoregulation (Jiang et al, 2004). Dermal mesenchymal cells contribute physical resilience by synthesizing extracellular matrix (ECM) components, such as collagen and elastin, and play a pivotal role in maintaining the homeostasis of epidermis, hair follicles, and vasculature structure via intercellular communication involving growth factors and cell adhesion (Di Carlo and Peduto, 2018; Ganier et al, 2022). Histological analyses and single-cell transcriptomes of the adult human skin have identified various subtypes of dermal mesenchymal cells, including papillary fibroblasts, reticular fibroblasts, pro-inflammatory fibroblasts, mesenchymal fibroblasts, dermal papilla cells, arrector pili muscle fibroblasts, pericytes, and mesenchymal stem cells (Tabib et al, 2018; Philippeos et al, 2018; Vorstandlechner et al, 2020; Ascensión et al, 2021; Bensa et al, 2023). Transplantation and lineage-tracing studies in murine models have revealed that dermal mesenchymal cells arise from two distinct fibroblast lineages: Lrig1- or Blimp1-expressing upper dermal fibroblasts and Dlk1-expressing lower dermal fibroblasts, each exhibiting unique contributions to cutaneous development and tissue regeneration (Driskell et al, 2013). The heterogeneity of dermal mesenchymal cells is modulated by factors, such as age, anatomical location, both intrinsic and extrinsic stressors, thereby influencing physiological processes including wound healing, fibrosis, atopic dermatitis, and aging (Solé-Boldo et al, 2020; Reynolds et al, 2021; He et al, 2020b). Interestingly, the potential for intervention in aberrant alterations of heterogeneity of dermal mesenchymal cell is being increasingly demonstrated, as exemplified by the suppression of ultraviolet-induced selective depletion of papillary fibroblasts through local cyclooxygenase 2 inhibition, and attenuation of aging-associated transcriptomic changes in dermal fibroblasts by caloric restriction (Ma et al, 2020; Salzer et al, 2018). Therefore, elucidating the functional characteristics of dermal mesenchymal cell subtypes and mechanisms that maintain homeostasis of the heterogeneity may significantly contribute to developing precision medical technologies targeting skin diseases and aging.

The identification and determination of functional characteristics of dermal mesenchymal cell heterogeneity have been

[1]ROHTO Pharmaceutical Co., Ltd., Ikuno-ku, Osaka 544-8666, Japan. [2]Institute of Advanced Biomedical Engineering and Science, Tokyo Women's Medical University (TWIns), 8-1 Kawada-cho, Shinjuku-ku, Tokyo 162-8666, Japan. [3]Human Biology Research Unit, Institute of Integrated Research, Institute of Science Tokyo, Bunkyo-ku, Tokyo 113-8510, Japan. ✉E-mail: skimura@rohto.co.jp

conducted through in vivo studies in mice and humans. However, animal experimentation requires careful consideration of inter-species differences with humans and faces increasing societal demands for a reduction (Bassi et al, 2021; Ferreira et al, 2022; Voelkl et al, 2020; Lynch and Watt, 2018). In contrast, human clinical studies are valuable for basic research and for validating efficacy; however, they are not suitable for large-scale screening of functional materials. To facilitate the societal application of findings regarding dermal mesenchymal cell heterogeneity, it is essential to develop in vitro research models that can recapitulate and assess cellular diversity and organ-level functionality. Human cell-based in vitro research models such as two-dimensional (2D)-tissue cultured cells, skin organoids, and human skin equivalents (HSEs) are being explored as preclinical research tools for avoiding animal experimentation(Marx, 2024). Although 2D tissue culture offers benefits for high-throughput screening, numerous subtypes of dermal mesenchymal cells, including papillary fibroblasts and pericytes, progressively lose their distinctive gene expression patterns and cellular morphology with successive passaging (Philippeos et al, 2018; Paquet-Fifield et al, 2009). Spheroid culture-based organoids, which allow for sophisticated in vitro reconstruction of skin organ structures, including hair follicles, vasculature, and nerves, and the cellular heterogeneity of their constituent components, hold significant potential for advancing the understanding of cell–cell interactions and for future integration into drug screening platforms (Hong et al, 2023; Toyoshima et al, 2012; Ibrahim and Richardson, 2017; Nebuloni et al, 2023). HSEs, distinguished by their capacity to fabricate tissues of customizable shapes at the centimeter scale, are already employed in the healthcare industry as commercially available platforms for tissue-level evaluation in vitro, encompassing drug safety, perme-ability, moisturizing barrier function, and mechanobiological aspects of skin physiology (Ali et al, 2015; Kimura et al, 2020; Bouwstra et al, 2021).

Basic HSEs, including those commercially available, comprise a fully stratified epidermal layer, along with a dermal equivalent formed from three-dimensional (3D) ECM such as a decellularized dermis or a Type 1 collagen seeded with dermal fibroblasts (Bell et al, 1979; Nusgens et al, 1984). The advancement of HSEs that relatively closely replicate the structure and functionality of native human skin is actively progressing by incorporating additional cell types, such as fibroblast subpopulations, vascular cells, immune cells, neural cells, and adipocytes, along with the integration of spheroid cultured tissues and application of 3D bioprinting technologies (Hofmann et al, 2023; Hosseini et al, 2022; Weng et al, 2021). The vascular network, which is indispensable for delivering oxygen and nutrients and regulating inflammatory responses, has been integrated into HSEs by incorporating vascular endothelial cells within the dermal compartment, transplanting endothelial cells into bioprinted vasculature-like structures, and implementing perfusion culture systems (Ponec et al, 2004; Black et al, 1998). These innovations have facilitated the application of HSEs in drug delivery research (Abaci et al, 2016). Although the development of HSEs that reconstruct the cellular heterogeneity of native skin remains relatively limited, recent single-cell transcrip-tomic analyses have demonstrated that HSEs recapitulate the principal in vivo epidermal keratinocyte subtypes, and encompass unique keratinocyte populations exhibiting aberrant signaling pathways associated with epithelial–mesenchymal transition.

Intriguingly, the appearance of these artificial cell subtypes is rescued by incorporating fibroblasts into dermal equivalents and xenografting onto immunodeficient mice, indicating that inter-cellular communication with keratinocytes and dermal mesench-ymal cells plays a crucial role in maintaining the homeostasis of epidermal cell heterogeneity (Stabell et al, 2023). In contrast, the heterogeneity of dermal mesenchymal cells of HSEs remains largely unexplored. To investigate the functional roles of distinct dermal mesenchymal cell subpopulations, HSEs have been constructed using papillary fibroblasts and pericytes derived from primary cultures of human dermis. These models have revealed that epithelialization is strongly influenced by intercellular communica-tion, including Wnt signaling and laminin subunit alpha 5 (LAMA5)-mediated pathways (Philippeos et al, 2018; Sriram et al, 2015; Paquet-Fifield et al, 2009; Mine et al, 2008). However, in constructing HSEs that replicate the heterogeneity of dermal mesenchymal cells, the strategy for isolating, culturing, and spatially arranging each cellular subpopulation is economically impractical.

In developing and regenerating different organs, cell-to-cell communication plays an essential role in inducing and maintaining differentiation of constituent cells. A specific subset of tissue-resident mesenchymal cells located close to blood vessels plays a key role in epidermal differentiation and ECM production (Paquet-Fifield et al, 2009; Ganier et al, 2022; Goss et al, 2021) Therefore, we hypothesized that by vascularizing an HSE and reproducing cell-to-cell communication on the basis of reconstructed perivascular niche, constructing an HSE that relatively faithfully replicates the in vivo characteristics of skin would be possible. In this study, we report an HSE that replicates the heterogeneity of human dermal mesenchymal cells, including pericytes, papillary fibroblasts, and reticular fibroblasts, by utilizing commercially available primary cells, specifically normal human epidermal keratinocytes (NHEKs), normal human dermal fibroblasts (NHDFs), and human umbilical vein endothelial cells (HUVECs). Additionally, we demonstrate that our HSEs serve as a valuable model for elucidating the role of dermal mesenchymal cell heterogeneity in preserving skin struc-tural integrity and functional homeostasis.

## Results

### Tricellular communication among epidermal keratinocytes, dermal fibroblasts, and vascular endothelial cells enhances the organization of skin and blood vessels

To investigate the role of communication among epidermal keratino-cytes, dermal fibroblasts, and vascular endothelial cells in skin tissue formation, we reconstructed seven types of HSEs, each incorporating one to three types of these constituent cells. The HSEs were named using the initials of the incorporated cell types. Specifically, HSEs composed of a single cell type, epidermal keratinocytes, dermal fibroblasts, or vascular endothelial cells, were referred to as the E, D, or V models, respectively. Models containing two of these cell types were designated as the DV, EV, and ED models, and the model incorporating all three cell types was referred to as the EDV model (Fig. 1A,B). All HSEs were constructed using a previously reported method for reproducing tensional homeostasis in a skin model

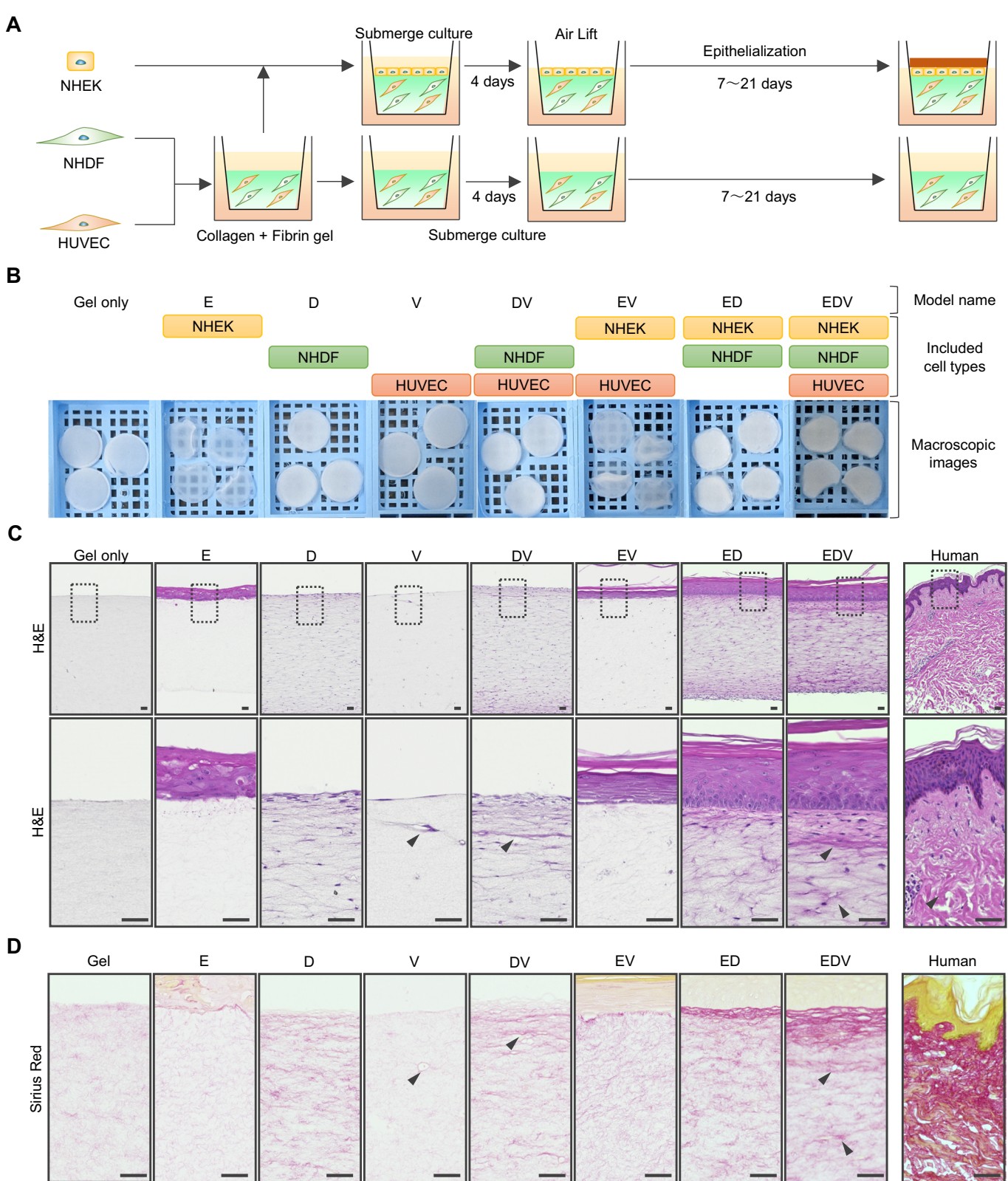

**Figure 1. Tricellular communication regulates skin and vascular organization.**

(A) Schematic of the construction method of Human Skin Equivalents (HSEs). (B) Macroscopic images of the HSEs. Scale bars, 1 mm. (C) H&E-stained images of human skin and the HSEs. Dotted box areas are enlarged in the lower panels. The arrowheads indicate the endothelial tube. Scale bars, 50 μm. (D) Sirius red-stained images of human skin and the HSEs. The arrowheads indicate the endothelial tube. The arrowheads indicate the endothelial tube. Scale bars, 50 μm.

(Kimura et al, 2020). After 14 d of air lift culture, the reconstructed tissue was 12 mm in diameter and 1–2 mm in thickness. The E and EV models were transparent and extremely soft (Fig. 1B). Hematoxylin and eosin (H&E) staining was used to visualize the regulation of skin and blood vessel organization through cell-to-cell communication (Fig. 1C). In the E and EV models, an abnormal epidermis displayed a disrupted basal cell layer formed, whereas normal epithelialization was observed in the ED and EDV models. Endothelial tubes in the dermis were observed only sparsely in the V and EV models; however, they were distributed throughout the skin in the DV and EDV models. In particular, the endothelial tubes in the EDV model tended to orient horizontally along the skin surface and increased in number with increasing distance from the epidermis. Sirius red staining revealed localized collagen fiber deposition (Fig. 1D). In the ED model, collagen fiber formation was confirmed at the epidermal–dermal junction because of the well-known interaction between the epidermis and dermis. In the DV and EDV models, the distribution of collagen deposition within the dermis was similar to that of the endothelial tubes. Interestingly, collagen deposition at the epidermal–dermal junction and around endothelial tubes was promoted in the EDV model compared to that in the ED and DV models.

The morphogenesis of the epidermis, endothelial tube, and dermis in each HSE was analyzed by using immunohistology. The results of epidermal marker staining demonstrated that the epidermis of the ED and EDV models exhibited a cytokeratin 5 (CK5)-positive basal layer, a claudin 1 (CLDN1)-positive basal to granular layer, filaggrin (FLG)-positive keratohyalin granules, and a cytokeratin 10 (CK10)-positive stratum corneum, closely resembling the epidermis of natural human skin (Fig. 2A). Furthermore, the number of Ki67-positive cells was highest in the EDV model and relatively low in the ED, EV, E models, and human skin (Fig. 2B).

Transepidermal water loss (TEWL) serves as an index for assessing the barrier function of viable skin. Based on a systematic review and meta-analysis, the estimated TEWL values (95% confidence interval) for individuals aged 18–64 years were 15.4 (13.9–17.0) g/m²/h for the right cheek, 6.5 (6.2–6.8) g/m²/h for the midvolar region of the right forearm, and 36.3 (29.5–43.1) g/m²/h for the right palm (Kottner et al, 2013). TEWL in the EDV model was 9.68 g/m²h, and water evaporation was suppressed compared to that in the ED model (Fig. 2C). These results indicate that dermal fibroblast–epidermal keratinocyte communication, that promotes epithelialization and barrier function is enhanced by vascular endothelial cells.

Immunostaining for the endothelial cell markers CD31 and von Willebrand factor (vWF) revealed active endothelial tube formation in the presence of both HUVECs and NHDFs (Fig. 2D). In the DV and EDV models, deposition of collagen types 1 and 4 was observed around the endothelial tubes, suggesting fibroblast–endothelial cell crosstalk stabilizes the endothelial tube structure through ECM synthesis (Fig. 2D). In the ED model, formation of type 1 collagen fiber in the dermis was stimulated around the epidermal–dermal junction, whereas in the EDV model, additional strong type 1

collagen expression originating from endothelial tubes was observed, resulting in the formation of collagen fibers that spread throughout the entire dermis. Microcirculation in natural human skin is organized into horizontal plexuses situated 1–1.5 mm below the skin surface (Braverman, 2000). Immunostaining for CD31 and type 1 collagen revealed that the blood vessels and collagen fiber in the EDV model were distributed in parallel to the epidermal plane, and mathematical assessment using 2D fast Fourier transform (2D-FFT) analysis corroborated this observation (Fig. 2e). The collagen network, which is arranged parallel to the skin surface, functions to withstand mechanical forces acting perpendicular to the skin (Kimura and Tsuji, 2021). Therefore, we evaluated the elasticity of HSEs, a physical indicator of skin aging and correlating factor of wrinkles and sagging, using the Cutometer (Ezure et al, 2009). The Cutometer is an instrument that employs negative pressure to elevate the skin surface, quantifying its mechanical properties by analyzing temporal variations in skin displacement. Notably, the R7 parameter which defined as the ratio of the immediate return height post-release of negative pressure to the maximum elevation during suction, exhibits a correlation with aging. Under the same measurement conditions, human skin displacement across ten suction cycles ranged from approximately 0.05 mm to 0.35 mm, with comparable trends observed in both the ED and EDV models (Fig. 2F) (Ohshima et al, 2013). Measurements were infeasible in HSEs other than the ED and EDV models, as these tissues could not endure the applied negative pressure and sustained structural damage. An increase in the R7 value was observed in the EDV model relative to the ED model (Fig. 2G).

Therefore, tricellular interactions among epidermal keratinocytes, dermal fibroblasts, and endothelial cells play critical roles in epidermal turnover, collagen synthesis, and angiogenesis and are sufficient to enhance tissue-scale skin functionality, including viscoelastic properties, and integrity of the epidermal barrier.

## Cell-to-cell communication in a 3D environment replicates dermal mesenchymal cell heterogeneity that is lost in 2D culture

To evaluate the impact of vascularization on fibroblast heterogeneity in the HSEs, we analyzed the spatial distribution of cells with positive markers for papillary fibroblasts (CD39 and fibroblast activation protein (FAP)), reticular fibroblasts (CD36 and CD90), and pericytes neural/glial antigen (NG2) and alpha-smooth muscle actin (αSMA) via immunohistochemistry (Fig. 3A,C). Additionally, the area ratios of expression regions for each dermal mesenchymal cell marker were quantitatively analyzed (Fig. 3B,D). Fibroblasts in the D model were positive for the pan-fibroblast marker vimentin; however, the expression of all markers in papillary and reticular fibroblasts and pericytes was barely detectable. CD39-positive cells were observed in the DV and EDV models, and FAP-positive cells were observed in the ED and EDV models. Notably, the areas positive for CD39 and FAP were increased in the EDV model

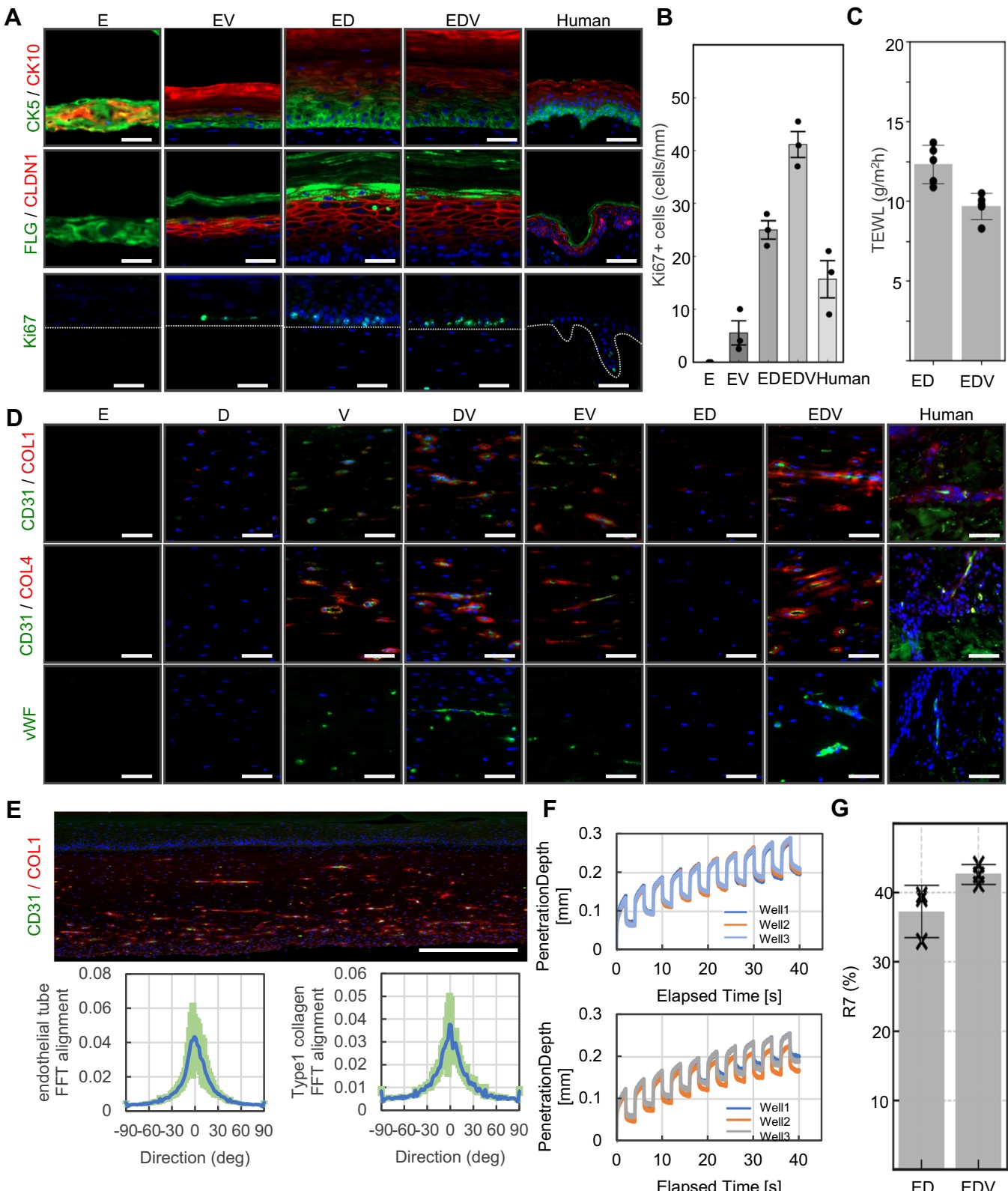

Figure 2. Tricellular communication enhances skin barrier function and elasticity by inducing self-organization.

(A) Immunohistochemical analyses of human skin and the HSEs for evaluating epithelialization. The dotted lines indicate the boundary between the epidermis and dermis. Scale bars, 50 μm. (B) Analysis of the Ki67-positive basal keratinocyte ratio in HSEs ($N = 3$, technical replicates). Data were presented as mean ± SEM to indicate the precision of the group mean for statistical comparison. (C) TEWL analysis of the ED and EDV models ($N = 5$, technical replicates), Data are presented as mean ± SD to reflect biological variability among independent samples. (D) Immunohistochemical analyses of human skin and HSEs for evaluating angiogenesis and dermal collagen synthesis. Scale bars, 50 μm. (E) Macroscopic image of CD31 and COL1 distribution (upper column) and FFT analysis (lower columns) of the alignment of CD31-positive endothelial tubes and COL1-positive type 1 collagen fibers in the EDV model ($N = 18$, technical replicates). Scale bars, 500 μm. (F) Skin deformation curves of the ED (upper panel) and EDV models (lower panel), ($N = 3$, technical replicates). (G) Evaluation of R7 factors of the ED and EDV models ($N = 3$, technical replicates). Data were presented as mean ± SD to reflect biological variability among independent samples.

compared to the other models. Interestingly, CD39-positive cells tended to localize around the endothelial tubes, whereas FAP-positive cells were observed mainly near the epidermal–dermal junction; similar patterns were also observed in natural human skin. CD36-positive cells were distributed in the DV and EDV models, and CD90-positive cells were sparsely distributed in the ED, and EDV models (Fig. 3A,B). NG2- and αSMA-positive pericytes were detected exclusively in the EDV model, and localized to the periphery of the lumen formed by vWF-positive vascular endothelial cells (Fig. 3C,D).

Immunohistochemical analysis revealed that the fibroblasts constituting the HSEs exhibited heterogeneity in terms of expression of the cell surface markers, depending on the constituent cell types. Therefore, we conducted single-cell RNA sequencing (scRNA-seq) to analyze the single-cell transcriptomes of dermal mesenchymal cell populations within the HSEs. We profiled three samples: the ED and EDV models cultured at the air–liquid interface for 14 d, a mixture of NHEKs, NHDFs, and HUVECs cultured separately under 2D conditions (referred to as 2D) (Fig. 4A). In an initial analysis, we obtained an overview of diverse cell populations by integrating cells from three samples. After quality filtering, a total of 23,911 cell profiles were successfully obtained across all conditions. A uniform manifold approximation and projection (UMAP) plot revealed three major cell populations and 20 distinct clusters (Fig. 4B, left). Based on the existing marker genes, we identified four cell types (keratinocytes, fibroblasts, vascular endothelial cells, and pericytes) for each cell culture (Fig. 4B, right). The UMAP plot illustrates specific impact of each culture condition on fibroblast population (Fig. EV1A). The major fibroblast clusters under 2D, #1, #6, #13, and #17 were relatively minor subpopulations in the ED and EDV models. The clusters #0, #2, #9, #10, and #11 were major in both the ED and EDV models; however, cluster #14, characterized by the expression of pericyte marker genes (ACTA2 and RGS5), was detected as a subpopulation unique to the EDV model (Fig. 4C). To test the hypothesis that fibroblast cluster formation and maintenance depend on interactions with endothelial cells or keratinocytes, we applied pseudotime trajectory analyses by monocle3 and CellChat analyses (Jin et al, 2024; Cao et al, 2019; Trapnell et al, 2014). First, we visualized the differentiation trajectories of fibroblast clusters distinguished on the UMAP plot (Fig. 4B). The analysis revealed that fibroblast clusters #6, #13, and #17, which were predominant in the two-dimensional culture environment, sequentially differentiated into clusters #1 (Y67-Y20) and #10 (Y20-Y76 and Y20-Y61) (Fig. EV1B). From cluster #10 fibroblasts, further differentiation into clusters #0 (Y76-Y25), #2 (Y76-Y54), #11 (Y76-Y1), and #9 (Y61-Y13) was observed (Fig. EV1B). To elucidate the

differentiation trajectory leading to cluster #14 pericytes, we reanalyzed the pseudotime trajectory analysis by increasing the number of center points used in the learn_graph function from 200 to 2000. This analysis indicated that cluster #9 fibroblasts, derived from clusters #10 and #11, further differentiated into cluster #14 pericytes (Y184-Y528) (Fig. 4D). To characterize the differentiation of each fibroblast and pericyte cluster and infer the underlying molecular mechanisms, we calculated Spearman's rank correlation coefficients between gene expression and pseudotime, selecting genes with absolute correlation values greater than 0.7. Only a small number of genes were identified along fibroblast-to-fibroblast differentiation trajectories, whereas 39 genes—including KDR (VEGFR2), EGFL6, FGFR3, and TGM2, which are associated with pericyte differentiation and vascular development—were extracted along the differentiation path from cluster #9 fibroblasts to cluster #14 pericytes (Dataset EV1). CellChat analysis further identified characteristic intercellular signaling between Cluster 14 fibroblasts and endothelial cells in Clusters #4 and #5, including vascular endothelial growth factor (VEGF)B–VEGF receptor 1 (VEGFR1), placental growth factor (PGF)–VEGFR1, laminin–dystroglycan, and laminin–integrin interactions (Fig. 4E). The ED and EDV models promoted the expression of collagen fiber-related genes such as COL1A1, COL3A1, COL4A1, COL4A2, and COL5A3, but suppressed the expression of elastic fiber-related genes, including ELN and FBN1, compared to that in the 2D conditions (Fig. 4F). This change in the transcriptome was relatively highly pronounced in the EDV model, suggesting the contribution of cluster #14.

## Integration of in vivo and HSE scRNA-seq datasets reveals distinct dermal fibroblast subpopulations

To investigate the biological significance of dermal mesenchymal cell heterogeneity of the HSEs, we referred to a published in vivo human skin scRNA-seq dataset and integrated it with our HSE datasets using the R Harmony data integration algorithm (Solé-Boldo et al, 2020), (GSE130973 (https://www.ncbi.nlm.nih.gov/geo/query/acc.cgi?acc=GSE130973), Data ref: Solé-Boldo et al, 2020). A UMAP plot revealed four major cell populations and 11 distinct clusters (Fig. 5A). To distinguish the results of this integrated analysis from previous in vitro data analysis, apostrophes were added to the cluster numbers in this analysis. The four major clusters were defined as keratinocytes, fibroblasts, vascular endothelial cells, and pericytes according to the expression patterns of marker genes (Fig. 5B). In the ED and EDV models, similar to in vivo skin, fibroblasts were composed of mainly clusters #0', #1', and #8'. In contrast, 2D-cultured fibroblasts contained very few cells from clusters #0' and #8', whereas clusters #2' and #11', which

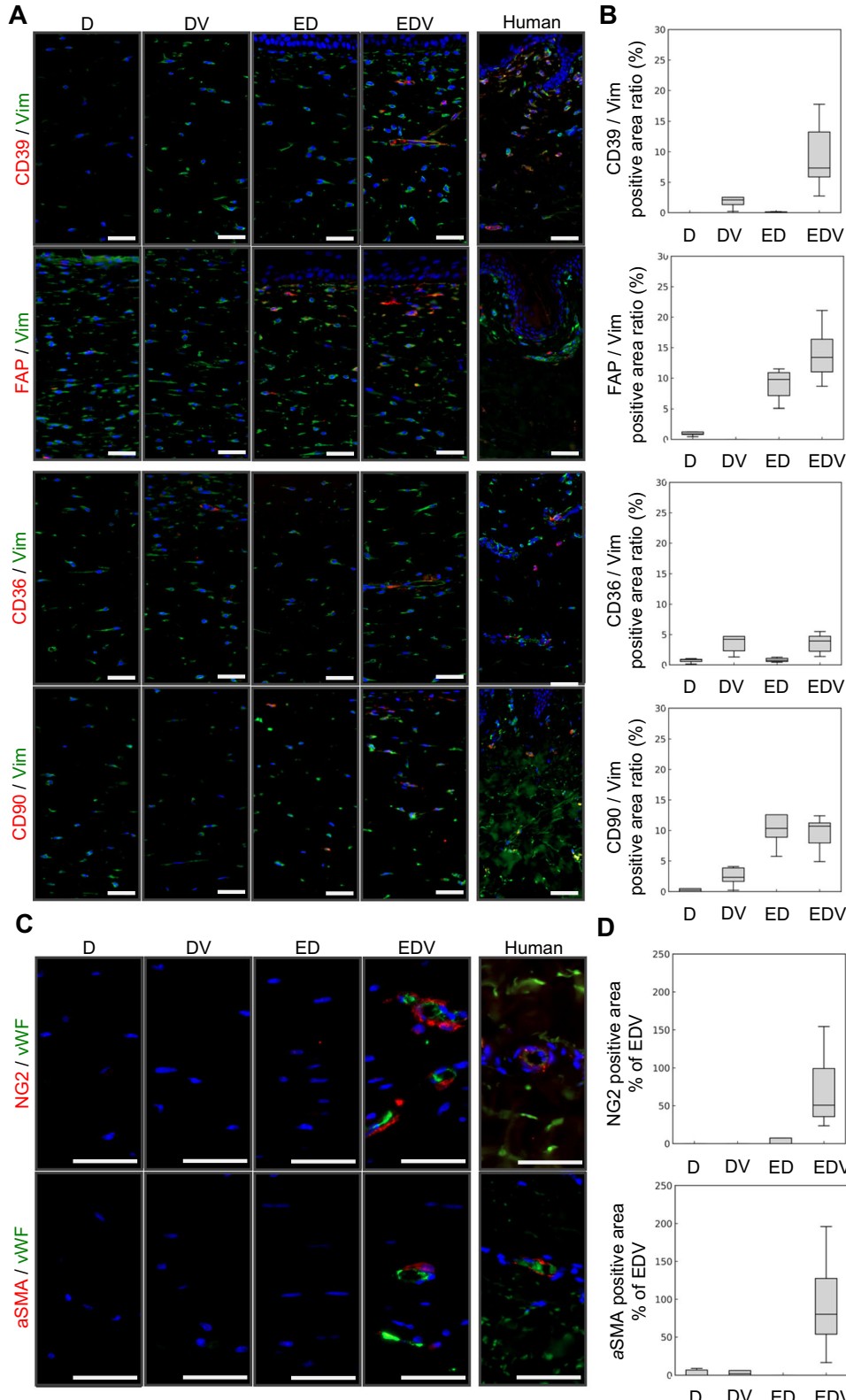

**Figure 3. Immunohistological analysis of dermal mesenchymal cell distribution in the HSEs and human skin.**

(A) Immunohistochemical analyses of ED, EDV and human skin samples to evaluate the presence and distribution of dermal mesenchymal cell subpopulations. Scale bars, 50 μm. (B) Quantitative analysis of the ratio of fibroblast marker-positive area relative to vimentin. Box plots depict the median as the center line, the interquartile range as the box, and whiskers extending to 1.5 × the interquartile range. Outliers were omitted. $N = 9$, technical replicates. (C) Immunohistochemical analysis of pericyte distribution. Scale bars, 50 μm. (D) Quantitative analysis of the ratio of pericyte marker-positive area. Box plots depict the median as the center line, the interquartile range as the box, and whiskers extending to 1.5 × the interquartile range. Outliers were omitted. ($N = 9$, technical replicates).

are rarely found in natural human skin, comprised their main subpopulations (Fig. 5C). Cell cycle scoring using the Seurat scRNA-seq analysis pipeline (ccSeurat) (Butler et al, 2018) (https://satijalab.org/seurat/v3.1/cell_cycle_vignette.html) revealed that cells in cluster #2', characteristic of 2D-cultured fibroblasts, were in the G2/M and S phases (Fig. 5D). Four main fibroblast subpopulations that could be spatially localized and showed differential secretory, mesenchymal and pro-inflammatory functional annotations have been previously defined (Solé-Boldo et al, 2020). We reanalyzed this dataset for identifying these four fibroblast subtypes and pericytes (Fig. EV2). We then plotted the cells corresponding to each subtype on the integrated UMAP plot (Fig. 5E). Next, we analyzed the clustering ratios within the integrated UMAP plot for each of the five natural human skin subpopulations of cells including secretory papillary, secretory reticular, pro-inflammatory, mesenchymal fibroblasts and pericytes (Fig. 5F). The results revealed that secretory papillary and pro-inflammatory fibroblasts corresponded to cluster #0'; secretory reticular fibroblasts corresponded to cluster #1'; mesenchymal fibroblasts corresponded to cluster #8'; and pericytes corresponded to cluster #6'. We also investigated the Gene Ontology (GO) terms enriched in cluster #0'; that represents the papillary fibroblast phenotype and cluster #1'; that represents the reticular fibroblast phenotype. In cluster #1'; GO terms related to collagen and elastic fiber formation were particularly enriched, suggesting that cells in cluster #1' exhibited a phenotype closely resembling that of reticular fibroblasts (Fig. 5G).

## Nutrient-poor culture conditions in the HSEs induce a skin barrier and elasticity disruption, which is rescued by ascorbic acid

Since blood vessels are central to maintaining organ homeostasis, vascular aging is hypothesized to be a fundamental upstream factor of organismal aging (Rastogi et al, 2021; Le Couteur and Lakatta, 2010). During skin aging, a reduction in blood vessels within the papillary layer and decline in their functions are thought to contribute to thinning of the epidermis and papillary dermis owing to impaired supply of nutrient and paracrine factors from vascular endothelial cells; however, experimental evidence for this phenomenon remains limited (Bonta et al, 2013; Braverman and Keh-Yen, 1981; Gunin et al, 2015). Therefore, to evaluate the impact of plasma-derived nutrient deprivation induced by peripheral vascular reduction, ED and EDV were cultured for 10 days in a nutrient-poor (NP) medium characterized by low FBS concentration and lack of basic fibroblast growth factor (bFGF) and ascorbic acid (AA) compared with the growth medium. Moreover, ascorbic acid, a well-evaluated antiaging material known for its antioxidant properties, ability to induce epithelialization, and role as a coenzyme in collagen synthesis (Masaki, 2010; Boo, 2022; Pullar

et al, 2017), was added to the NP medium for evaluating its impact on HSE organization and skin functionality. H&E staining and immunohistological analysis revealed that skin tissue, including the epidermis, dermis, and endothelial tubes with pericytes, was reconstructed even under NP culture conditions (Fig. 6A,B). However, the number of Ki67-positive cells in the epidermal basal cell layer was lower in the EDV model than in the ED model (Fig. 6C). When 500 μM AA was added, thickening of the CK5- and/or CK10- positive epidermal layer and promotion of type 1 collagen deposition were observed in both the ED and EDV models (Fig. 6B,D). AA increased the number of Ki67-positive cells in both the ED and EDV models; however, the EDV model showed a significantly relatively high number in the number of Ki67-positive cells (Fig. 6C). In both the ED and EDV models, ascorbic acid reduced TEWL and enhanced skin barrier function (Fig. 6E). A 2D-FFT analysis revealed that, NP culture conditions with AA resulted in horizontal orientation of CD31-positive vascular endothelial cells and the surrounding type 1 collagen relative to the epidermal layer, compared to that of the NP alone culture conditions (Fig. 6F). Regardless of the presence or absence of ascorbic acid, NG2- and αSMA-positive pericytes were consistently observed surrounding the endothelial tubes, with no detectable differences in the positive areas of these markers (Fig. 6B,G). Concerning the expression patterns of papillary and reticular fibroblast markers, the FAP-positive area was reduced upon ascorbic acid treatment, whereas CD39, CD36, and CD90 showed no significant changes. Across all culture conditions, the proportions of cells positive for the senescence markers p16 and p21 remained comparable, showing no statistically significant differences (Fig. EV3). Although no changes were detected in the ED model, skin elasticity was increased by AA in the EDV model (Fig. 6H). To explore the mechanism underlying the enhanced responsiveness of the EDV model to ascorbic acid compared with the ED model, we analyzed scRNA-seq data to compare the expression levels of ascorbic acid and dehydroascorbic acid transporter genes among fibroblast subpopulations. Cluster 14, characterized by high expression of pericyte markers, showed relatively elevated expression levels of *SLC2A1 (GLUT1)* and *SLC2A3 (GLUT3)* (Fig. EV4). These findings suggest that altered dehydroascorbic acid uptake by pericytes may contribute to the increased responsiveness of the EDV model to ascorbic acid.

In summary, NP culture conditions enabled in vitro replication of epidermal thinning, barrier disruption, reduction in dermal elasticity with imbalanced collagen synthesis and degradation, and disordered vascular orientation. Treatment with ascorbic acid ameliorated these impairments in skin function, with the extent of improvement being significantly greater in the EDV model than in the ED model. These findings demonstrate that our HSEs serve as a valuable tool for evaluating the efficacy of functional molecules, and suggest that vascular endothelial cells and pericytes play

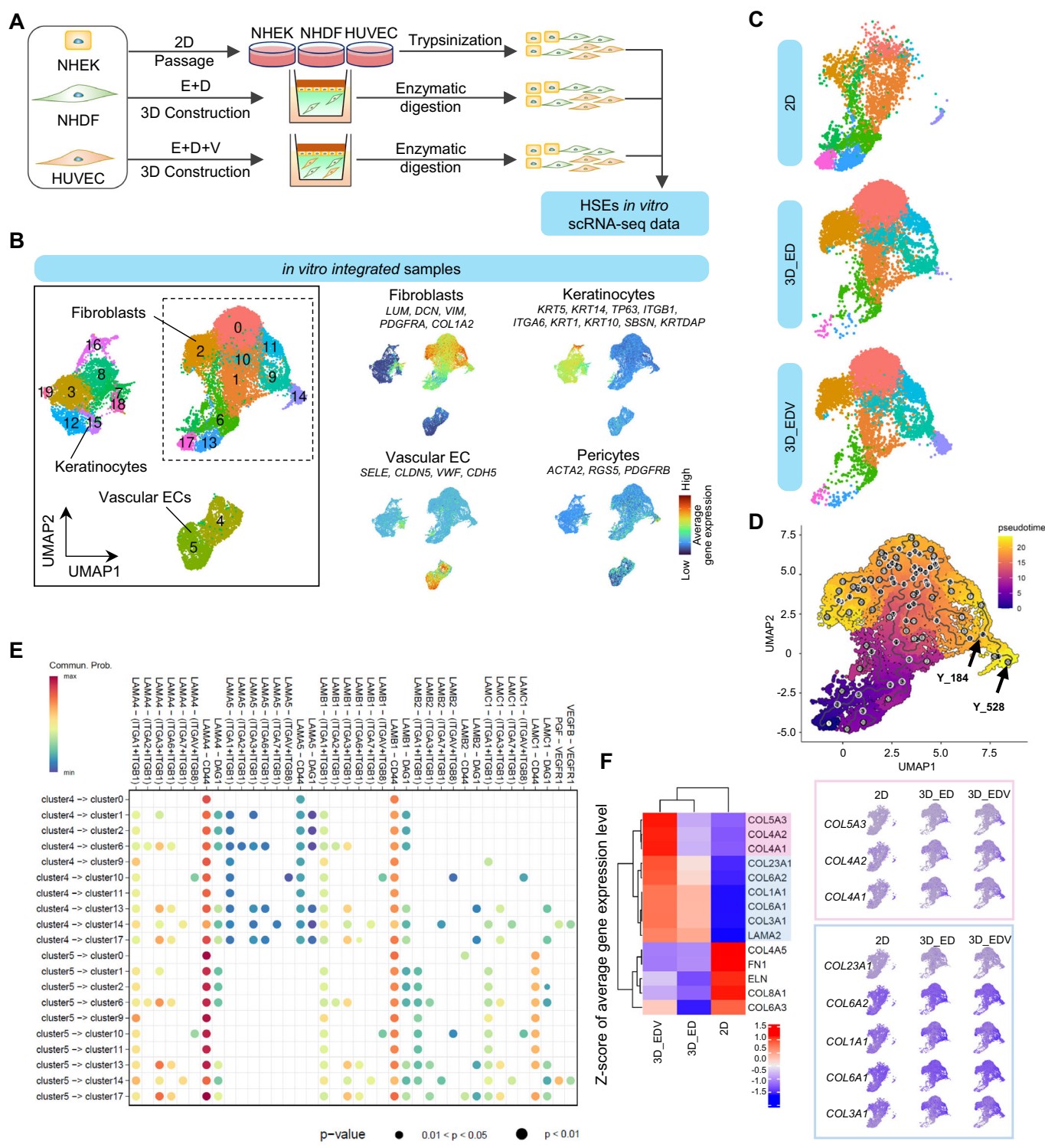

**Figure 4. The 3D culture environment and cell-to-cell communication replicate dermal mesenchymal cell heterogeneity in vitro.**

(A) Illustration of the workflow for sampling and analysis of the scRNA-seq data. (B) UMAP plot of the scRNA-seq data for in vitro HSEs; the average expression of four cell type markers was projected onto the UMAP plot to identify all cell population. (C) Distribution of fibroblasts and pericytes under each of the three conditions. (D) Pseudotime trajectory visualization of cells within the fibroblasts cluster with multiple annotated nodes. Color gradient represents pseudotime values, indicating progression along the differentiation trajectory. (E) Cellular communication from vascular endothelial cells (ECs; Clusters #4 and #5) to pericytes in Cluster #14 and fibroblasts in Cluster #9. Statistical significance was assessed using a permutation test implemented in CellChat (p < 0.05). (F) Heatmap showing the Z-score normalized average expression level of ECM-related genes across all fibroblasts under different conditions.

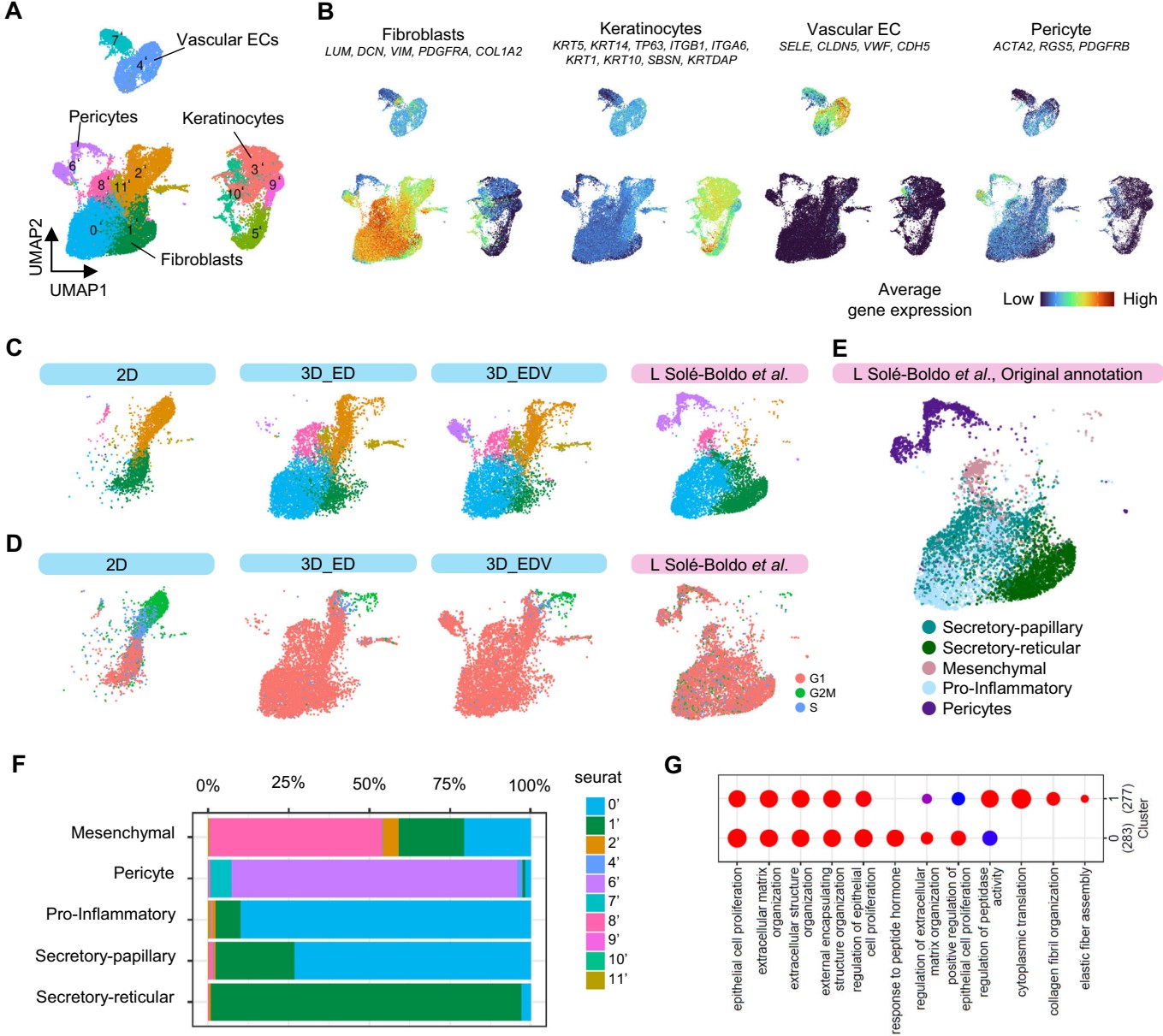

**Figure 5. Biological significance of fibroblast heterogeneity in HSEs through integrated analysis of human scRNA-seq data.**

In vivo skin single-cell analysis was carried out via methods described in previous studies(Solé-Boldo et al, 2020), (GSE130973, Data ref: Solé-Boldo et al, 2020). (A) UMAP plot of the scRNA-seq data for integrating HSEs and in vivo skin data. (B) Average expression of established cell type markers was projected on the UMAP plot to identify cell populations. Red indicates maximum gene expression, whereas blue indicates low or no expression of a particular set of genes in log-normalized UMI counts. (C) UMAP plot integrating the scRNA-seq data of HSEs and in vivo skin, showing the distribution of fibroblasts and pericytes across the three in vitro and in vivo conditions. (D) Cell cycle status of fibroblasts under each condition. (E) UMAP plot showing the correspondence of the four fibroblast subpopulations and pericytes, as previously reported11. (F) Proportion of cells from each subpopulation as previously defined 11 and their clustering into specific clusters, as shown in Fig. 5E. (G) Enriched GO terms in clusters #0' and #1'.

critical roles in the underlying mechanisms of action of these molecules.

# Discussion

In this study, we demonstrated that the heterogeneity of dermal mesenchymal cells, such as papillary and reticular fibroblasts and

pericytes, in natural human skin can be partially reproduced by reconstructing an HSE using NHEKs, NHDFs, and HUVECs. In particular, the EDV model containing a pericyte-like subpopulation improved skin and vascular tissue structure, barrier function, and elasticity, enabling functional evaluation. These results elucidate the role of vascular cells in developing the heterogeneity of dermal mesenchymal cells and demonstrate that the incorporation of vasculature into HSEs is an effective approach for recapitulating

nutrient supply and inflammatory responses, as traditionally considered, and achieving a relatively highly advanced reconstruction of the skin transcriptome and functional properties. While the reconstructed pericytes, secretory-reticular fibroblasts, and mesenchymal fibroblasts in the HSE exhibited characteristics relatively similar to their in vivo subpopulations, it should be noted that the model failed to distinguish between secretory-papillary fibroblasts and pro-inflammatory fibroblasts, indicating that the heterogeneity of dermal mesenchymal cell is only partially recapitulated (Fig. 5F). This limitation is likely attributable to the use of HUVECs as the vascular endothelial cell source which do not fully reproduce the phenotypic and functional properties of skin-specific microvascular endothelium. The absence of immune and adipose cells may also underlie the incomplete recapitulation of dermal mesenchymal cell heterogeneity by restricting the spectrum of fibroblast intercellular communication. Our HSE, which partially reproduces fibroblast heterogeneity using stably available primary culture cells, provides a model alternative to animal experiments, which can be used to analyze the mechanisms controlling skin physiological functions based on cell-to-cell communication.

Papillary fibroblasts play important roles in wound healing functions such as re-epithelialization and collagen synthesis. Furthermore, papillary fibroblasts derived from elderly donors have reduced ability of proliferation and low secretion levels of keratinocyte growth factor, suggesting that this effect may involve loss of the papillary dermis, and elasticity, and deformation of skin texture owing to aging (Sriram et al, 2015; Mine et al, 2008). In fact, recent scRNA-seq analyses of human skin have revealed changes in fibroblast subpopulations in the skin owing to aging and disease, suggesting the possibility of developing new therapeutic strategies by controlling fibroblast heterogeneity (Solé-Boldo et al, 2020; Tabib et al, 2018; He et al, 2020a). Epidermal Wnt signals are involved in inducing differentiation of the papillary fibroblast lineage, and intervention in dermal fibroblast subpopulations controls skin morphogenesis and functionality in animals (Driskell et al, 2013). However, owing to the lack of appropriate in vitro models and ethical restrictions to clinical in vivo studies, the prospects of applying in vitro findings of human fibroblast heterogeneity to in vivo health care are unclear. Indeed, Philippeos et al demonstrated that while CD39, CD90, and CD36 are detectable in primary CD31-CD45-Ecad- dermal cells, the expression of CD39 is lost after a single passage. In contrast, CD90 and CD36 remain detectable for up to four passages (Philippeos et al, 2018). These findings underscore the impact of in vitro culture on the depletion of fibroblast marker expression. Since we employed NHDFs that had undergone four to five passages for HSE reconstruction, it is reasonable to assume that these cells had already lost specific fibroblast subpopulations, including CD39[+] cells. Consistent with this, our scRNA-seq analysis revealed that most fibroblasts cultured in 2D formed an artificial population comprising cells in the S and G2M phases, along with secretory-reticular fibroblasts (Fig. 5C,D). Additionally, immunohistochemical analysis confirmed a near-complete absence of CD39[+], CD90[+], FAP[+], NG2[+], and αSMA[+] cells in the dermis of both D and DV models, further indicating that serial passaging significantly reduces the expression of markers associated with papillary fibroblasts, reticular fibroblasts, and pericytes (Fig. 3). Interestingly, cells positive for FAP, a marker for papillary fibroblasts, were present near the epidermal–dermal junction in both the ED and EDV

models; however, CD39-positive cells were observed only in the EDV model. This observation suggests that the keratinocyte–fibroblast communication is required for the induction and maintenance of papillary fibroblast differentiation, and that some subpopulations, such as CD39-positive cells, may require interaction with vascular endothelial cells. However, it is well established that excessive passaging of adipose-derived mesenchymal cells induces cellular senescence, leading to diminished proliferative and differentiation capacities. Therefore, the partial reconstitution of dermal mesenchymal cell heterogeneity observed in the EDV model may also be influenced by the passage number and senescence of fibroblasts, necessitating further validation for the development of a stable model. By elucidating the mechanisms controlling human fibroblast heterogeneity via cell tracking and spatiotemporal gene expression control techniques, our HSE may contribute to developing cell therapies for wound healing and aging.

In addition to papillary and reticular fibroblasts, perivascular mesenchymal cells, such as dermal stem cells and pericytes, play important roles in skin morphogenesis and function. Pericytes play an important role as the major source of ECM production during tissue regeneration in several organs (Kramann et al, 2015; Greenhalgh et al, 2013). HSEs reconstructed using human pericytes and fibroblasts separated by flow cytometry have been reported to promote the proliferation of epidermal basal cells through LAMA5 expression by pericytes (Paquet-Fifield et al, 2009). In our EDV model, epithelialization and dermal ECM synthesis were promoted with the emergence of a pericyte-like population. However, our HSE is unique in that it does not require fresh human skin and allows the construction of a model containing NG2- and αSMA-positive pericytes in a relatively simple process using commercially available subcultured cells. Pericytes are a heterogeneous cell population in vivo, and pericytes from the same tissue may originate from multiple sources (Dias Moura Prazeres et al, 2017; Goss et al, 2021). Integrative analysis of scRNA-seq data from natural human skin and our HSEs revealed that although the pericyte-like population in HSEs clustered with human pericytes within the same subpopulation, their distributions on the UMAP plot only partially overlapped. scRNA-seq analysis suggested that the pericyte-like Cluster #14 cells induced in the EDV model are derived from fibroblasts. This cluster exhibited intercellular interactions with endothelial cells mediated by signaling pathways such as *VEGFB–VEGFR1*, *PGF–VEGFR1*, and adhesion via laminin– and integrin-related signals, and recapitulated gene expression changes associated with pericyte differentiation during the transition from angiogenesis to vascular maturation, including *KDR (VEGFR2)* (Greenberg et al, 2008; Huang, 2020; Bergers and Song, 2005). These results suggest that the pericytes in our HSE partially replicate certain populations and functions of human pericytes, although the origin and function of this cell lineage and its mechanisms remain unclear. Cell tracking and ablation experiments or inhibitor experiments targeting VEGF signaling may contribute to addressing these gaps.

In addition to replicating the heterogeneity of dermal mesenchymal cells, the EDV model improved dermal elasticity and skin barrier function. This improvement in dermal elasticity is attributed to the functionality of active ECM synthesis by pericytes, compared to that by other subpopulations, and the autonomous recapitulation of oriented collagen fiber formation along

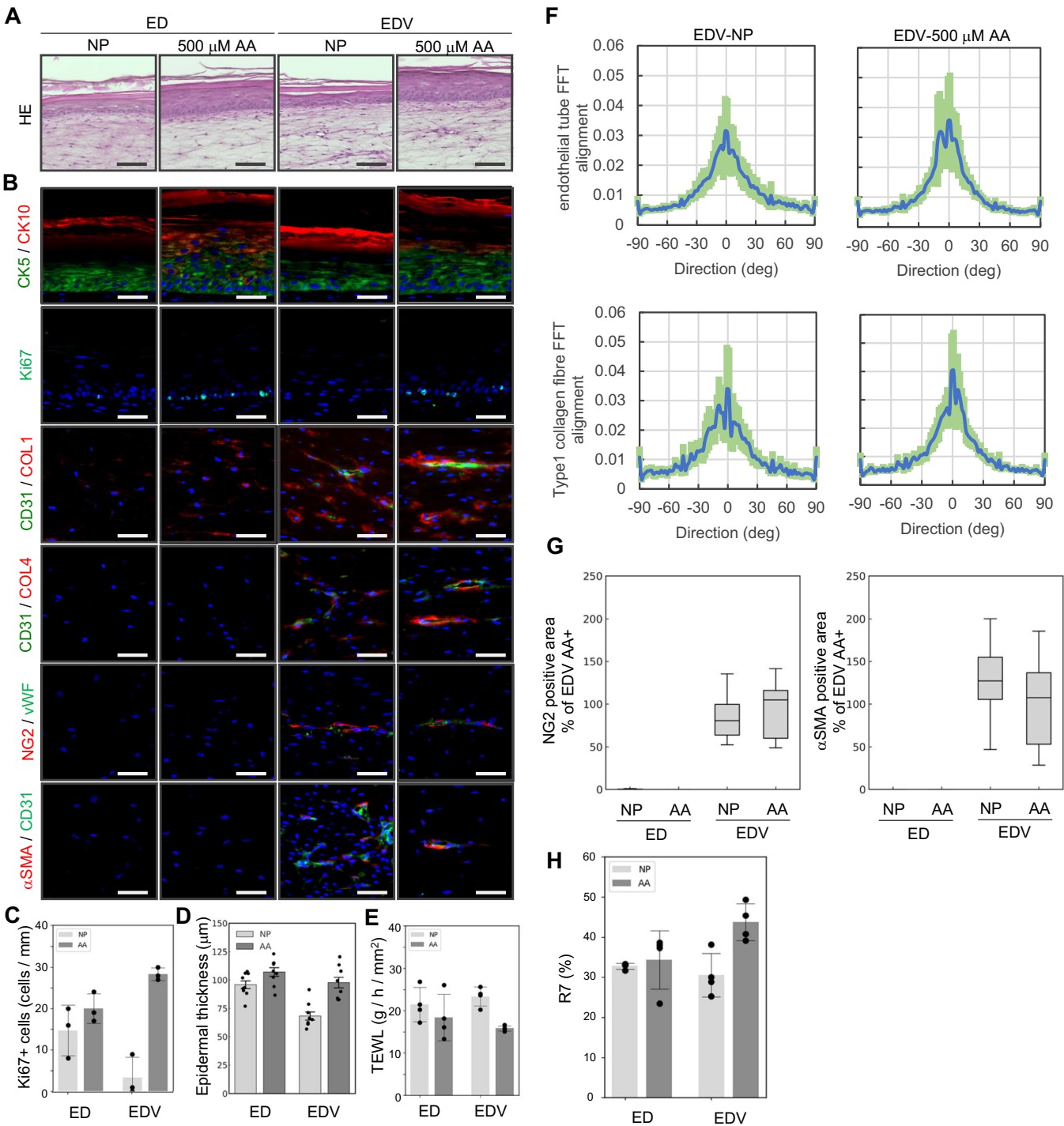

**Figure 6. NP culture conditions in the HSEs induce a senescent phenotype, which is suppressed by AA.**

(A) H&E-stained images of skin tissue organization induced by AA supplementation in the HSEs cultured under NP conditions. Scale bar, 100 μm. (B) Immunohistochemical analysis of the HSEs for visualizing the impact of NP conditions and ascorbic acid on re-epithelialization, dermal ECM synthesis, and angiogenesis. Scale bar, 50 μm. (C) Analysis of the Ki67-positive basal keratinocyte ratio in the HSEs. Data were presented as mean ± SEM to indicate the precision of the group mean for statistical comparison. (N = 3, technical replicates, two-tailed Student's t-tests). (D) Analysis of the epidermal thickness. Data were presented as mean ± SD to reflect biological variability among independent samples. N = 9, technical replicates. (E) TEWL analysis of the ED and EDV models. N = 4, technical replicates. Data were presented as mean ± SD to reflect biological variability among independent samples. (F) A 2D-FFT analysis (lower columns) of the alignment of CD31-positive endothelial tubes and COL1-positive type 1 collagen fibers in the EDV model (N = 18, technical replicates). (G) Quantitative analysis of the ratio of pericyte marker-positive area. Box plots depict the median as the center line, the interquartile range as the box, and whiskers extending to 1.5 × the interquartile range. Outliers were omitted. N = 9, technical replicates. (H) Evaluation of R7 factors of the ED and EDV models (N = 4, technical replicates, error bars represent standard deviation).

endothelial tubes. Furthermore, culturing under NP conditions enabled the HSE model to induce the aged skin-related phenotypes, including epidermal barrier dysfunction owing to reduced proliferation of basal keratinocytes, diminished dermal elasticity owing to decreased collagen synthesis, and loss of vascular orientation. In natural human skin, oxidative stress and dermal ECM degradation cause vascular atrophy and reduce vascular density, leading to depletion of nutrient and oxygen supplies to the papillary dermis and epidermis, which contributes to aging. Our findings support this hypothesis. In our NP HSEs, a reduction in vascular density was not observed, which is likely attributed to VEGF in the medium continuously inducing angiogenesis. AA, which functions as both an antioxidant and a coenzyme for 2-oxoglutarate-dependent dioxygenases, is an effective skin–antiaging agent. However, the effects of AA on the vascular endothelium and perivascular mesenchymal cells and its impact on skin mechanical properties remain unclear. Our findings revealed that ascorbic acid increased collagen synthesis in both the ED and EDV models, but improved skin viscoelasticity only in the EDV model, indicating that vascular endothelial cells and pericytes play crucial roles in regulating skin elasticity and serve as target cells for AA. Although the underlying molecular mechanism remains unclear, the elevated expression of *SLC2A3 (GLUT3)* observed in HSE pericytes suggests a potential involvement of dehydroascorbic acid uptake and metabolism. Further investigation, such as single-cell level analyses of intracellular ascorbic acid concentrations, will be required to elucidate this mechanism. In summary, our model emphasizes the importance of vascular cells in the study of skin care and reconstruction.

In conclusion, we reconstructed HSEs that reproduced skin fibroblast heterogeneity, tissue structure, and physiological function in natural human skin. Further studies, such as elucidating the mechanisms controlling fibroblast heterogeneity, evaluating the impact of each cell population on skin morphogenesis, and identifying drugs that can control the decrease or increase in specific fibroblast populations, may provide new insights into the fields of basic science, personalized medicine, and health care. Our model may help avoid animal experiments and contribute to the sustainable growth of the health care industry as a versatile tool for evaluating skin functionality as an alternative to clinical research. HSEs that highly replicate skin cell heterogeneity and function may be developed by improving their vascular structure through perfusion culture and providing cell-to-cell communication via organoids such as the IOS and nervous systems.

# Methods

### Reagents and tools table

| Reagent/resource | Reference or source | Identifier or catalog number |
|---|---|---|
| **Experimental models** | | |
| Normal human epidermal keratinocytes (NHEKs) | Kurabo | KK-4109 |
| Normal human dermal fibroblasts (NHDFs) | Kurabo | KF-4109 |

| Reagent/resource | Reference or source | Identifier or catalog number |
|---|---|---|
| Human umbilical vein endothelial cells (HUVECs) | Kurabo | KE-4109 |
| **Recombinant DNA** | | |
| None | None | None |
| **Antibodies** | | |
| Anti-Cytokeratin 5 antibody | Abcam | Ab52635 |
| Anti-Cytokeratin 10 antibody | Abcam | Ab9026 |
| Anti-Claudin 1 antibody | Abcam | ab211737 |
| Filaggrin Monoclonal Antibody (FLG01) | Thermo Fisher Scientific | MA5-13440 |
| Anti-Ki67 antibody | Abcam | ab156956 |
| Anti-Collagen I antibody | Abcam | ab34710 |
| Anti-Collagen IV antibody | Abcam | ab6311 |
| Anti-CD31 antibody | Abcam | ab9498 |
| Anti-CD31 antibody | Abcam | ab28364 |
| Anti-Von Willebrand Factor antibody | Abcam | ab778 |
| anti-Collagen Type 1 (3/4 fragment), pAb | AdipoGen | AG-25T-0113 |
| Anti-alpha smooth muscle Actin | Abcam | ab5694 |
| Anti-NG2 antibody | Abcam | ab139406 |
| Anti-Fibroblast activation protein, alpha antibody | Abcam | ab207178 |
| Anti-CD36 antibody | Abcam | ab252923 |
| Anti-CD39 antibody | Abcam | ab223842 |
| Anti-CD90/Thy1 antibody | Abcam | ab92574 |
| Anti-Vimentin antibody | Abcam | ab8978 |
| Anti-CDKN2A/p16INK4a antibody [EPR1473] - C-terminal | Abcam | ab108349 |
| p21 Waf1/Cip1 (12D1) Rabbit mAb | Cell Signaling | #2947 |
| Alexa Fluor® 594 donkey anti-rabbit IgG | Thermo Fisher Scientific | A21207 |
| Alexa Fluor® 594 donkey anti-mouse IgG (H + L) antibody | Thermo Fisher Scientific | A21203 |
| Alexa Fluor® 488 donkey anti-rabbit IgG (H + L) | Thermo Fisher Scientific | A21206 |
| Alexa Fluor® 488 donkey anti-mouse IgG (H + L) | Thermo Fisher Scientific | A21202 |
| **Oligonucleotides and other sequence-based reagents** | | |
| None | None | None |
| **Chemicals, enzymes and other reagents** | | |
| Dulbecco's modified Eagle's medium | Thermo Fisher Scientific | 11995-073 |
| 100 units ml$^{-1}$ penicillin, and 100 µg ml$^{-1}$ streptomycin | Thermo Fisher Scientific | 15140122 |
| HuMedia-KG2 | Kurabo | KK-2150S |
| HuMedia-EG2 | Kurabo | KE-2150S |

| Reagent/resource | Reference or source | Identifier or catalog number |
|---|---|---|
| Native Collagen Acidic Solution I-AC 5 mg/mL | KOKEN | IAC-50 |
| HEPES | Dojindo | 340-08233 |
| NaHCO₃ | FUJIFILM Wako Pure Chemical Corporation | 191-01305 |
| Insulin | FUJIFILM Wako Pure Chemical Corporation | 099-06473 |
| Ascorbic acid | FUJIFILM Wako Pure Chemical Corporation | 016-04805 |
| Hydrocortisone | FUJIFILM Wako Pure Chemical Corporation | 088-02483 |
| VEGF 165 Human Recombinant | PeproTech | 100-20-100µg |
| Trypsin/EDTA solution | Kurabo | HK-3120 |
| Trypsin-neutralizing solution | Kurabo | HK-3220 |
| Collagenase type 1 | Worthington Biochemical Corp | LS004196 |
| Thrombin from bovine plasma | Sigma | T4648-1KU |
| **Software** | | |
| ImageJ FIJI | https://imagej.net/ij/ | |
| BellCurve for Excel | Social Survey Research Information | |
| Seurat 5. 3. 0 | https://satijalab.org/seurat/ | |
| monocle3 1. 4. 26 | https://cole-trapnell-lab.github.io/monocle3/ | |
| dplyr 1. 1. 4 | https://dplyr.tidyverse.org/ | |
| CellChat 1. 6. 1 | https://github.com/jinworks/CellChat | |
| patchwork 1. 2. 0 | https://patchwork.data-imaginist.com/ | |
| harmony 1. 2. 4 | https://github.com/immunogenomics/harmony | |
| **Other** | | |
| AxioScan. Z1 | ZEISS | |
| LSM900 | ZEISS | |
| Cutometer MPA 580 | Courage & Khazaka electronic | |
| Tewameter TM HEX | Courage & Khazaka electronic | |

## Cell culture

Normal human epidermal keratinocytes (NHEKs) were purchased from KURABO Industries Ltd. (Osaka, Japan) and were maintained in HuMedia-KG2 (KURABO). Normal human dermal fibroblasts (NHDFs) were purchased from KURABO and maintained in Dulbecco's modified Eagle's medium (Thermo Fisher Scientific, MA, USA) supplemented with 10% fetal bovine serum (Corning, NY, USA), 100 units ml⁻¹ penicillin, and 100 µg ml⁻¹ streptomycin

(Thermo Fisher Scientific). Human umbilical vein endothelial cells (HUVECs) were purchased from KURABO and maintained in HuMedia-EG2 (KURABO). All cells were seeded into 150-mm dishes and grown in a humidified atmosphere of 5% $CO_2$ at 37 °C. For all cell types, passage 4 or 5 cells were utilized for the reconstitution of human skin equivalents (HSE).

## Reconstruction of the tension homeostatic skin (THS) model

The THS model was generated via the following procedure(Kimura et al, 2020). A total of $24 \times 10^5$ NHDFs and $30 \times 10^5$ HUVECs were resuspended in 2.4 ml of 1.25 mg ml⁻¹ bovine dermis-derived native collagen solution (KOKEN CO. Ltd., Tokyo, Japan) that was mixed with 1× DMEM (Thermo Fisher Scientific), 15 mM HEPES (Dojindo, Kumamoto, Japan), 10 mM $NaHCO_3$ (FUJIFILM Wako Pure Chemical Corporation), and 5 mg ml⁻¹ fibrinogen from bovine plasma type I-S (Merck KGaA, Darmstadt, Germany); then, the cells were plated in a high-density translucent-membrane cell culture insert (0.4-µm pore size) (Corning, NY, USA) in six-well plates. After solidification, the dermal equivalents were held in place by a Snapwell culture insert (Corning). NHEKs were seeded at a density of $1.1 \times 10^6$ cells/well on fixed dermal equivalents, and they were cultured in growth medium under submersion conditions for 4 days in a humidified atmosphere at 37 °C with 5% $CO_2$ and 12.5% $O_2$. After maintenance in a submersion culture, the HSEs were exposed to the air–liquid interface in a humidified atmosphere at 37 °C with 5% $CO_2$. The growth medium for culturing HSEs consisted of DMEM supplemented with 10% FBS, 1% penicillin/streptomycin, 5 µg ml⁻¹ insulin (FUJIFILM Wako Pure Chemical Corporation), 500 µM ascorbic acid (FUJIFILM Wako Pure Chemical Corporation), 10 ng ml⁻¹ bFGF (PeproTech, NJ, USA), 1 µM hydrocortisone (FUJIFILM Wako Pure Chemical Corporation), and 5 nM VEGF 165 Human Recombinant (PeproTech) and was replaced with fresh medium every 48 to 72 h.

## H&E staining and immunohistochemistry

For histological analysis, skin equivalents from three independent experiments were fixed with 4% formaldehyde and embedded in paraffin or Tissue-Tek OCT compound (Sakura Finetek Japan Co., Ltd., Tokyo, Japan). Hematoxylin-eosin staining was performed on paraffin sections (10 µm thick). The stained sections were observed via AxioScan.Z1 (Carl Zeiss, Oberkochen, Germany). For fluorescence immunohistochemistry, frozen sections (10 and 50 µm) and paraffin sections (10 µm) were prepared and stained as previously described (Toyoshima et al, 2012). Details of the primary and secondary antibodies and associated epitope recovery methods are included in Table 1. All fluorescence microscopy images were captured with an AxioScan.Z1 or LSM900 confocal microscope (Carl Zeiss).

## Quantitative analysis of epidermal cell proliferation

The proliferation rate of epidermal cells was quantified by counting keratinocytes with Ki67-positive nuclei located at the basal or immediately suprabasal epidermis. The measurement area was a total of three images obtained by randomly acquiring three z-stack images of $1000 \times 340 \times 10$ µm from HSE tissues that were derived from three different wells.

**Table 1.  Primary and secondary antibodies.**

| Antibodies | Source | Cat# | Dilution ratio |
|---|---|---|---|
| Primary | | | |
| Anti-cytokeratin 5 antibody | Abcam | Ab52635 | 1/200 |
| Anti-cytokeratin 10 antibody | Abcam | Ab9026 | 1/200 |
| Anti-claudin 1 antibody | Abcam | ab211737 | 1/250 |
| Filaggrin monoclonal Antibody (FLG01) | Thermo Fisher Scientific | MA5-13440 | 1/50 |
| Anti-Ki67 antibody | Abcam | ab156956 | 1/150 |
| Anti-collagen I antibody | Abcam | ab34710 | 1/200 |
| Anti-collagen IV antibody | Abcam | ab6311 | 1/250 |
| Anti-CD31 antibody | Abcam | ab9498 | 1/1000 |
| Anti-CD31 antibody | Abcam | ab28364 | 1/50 |
| Anti-Von Willebrand Factor antibody | Abcam | ab778 | 1/50 |
| Anti-collagen type 1 (3/4 fragment), pAb | AdipoGen | AG-25T-0113 | 1/50 |
| Anti-alpha smooth muscle Actin | Abcam | ab5694 | 1/100 |
| Anti-NG2 antibody | Abcam | ab139406 | 1/250 |
| Anti-fibroblast activation protein, alpha antibody | Abcam | ab207178 | 1/250 |
| Anti-CD36 antibody | Abcam | ab252923 | 1/250 |
| Anti-CD39 antibody | Abcam | ab223842 | 1/1000 |
| Anti-CD90/Thy1 antibody | Abcam | ab92574 | 1/200 |
| Anti-Vimentin antibody | Abcam | ab8978 | 1/500 |
| Anti-CDKN2A/p16INK4a antibody [EPR1473] - C-terminal | Abcam | ab108349 | 1/100 |
| p21 Waf1/Cip1 (12D1) Rabbit mAb | Cell Signaling | #2947 | 1/50 |
| Secondary | | | |
| Alexa Fluor® 594 donkey anti-rabbit IgG | Thermo Fisher Scientific | A21207 | 1/500 |
| Alexa Fluor® 594 donkey anti-mouse IgG (H + L) antibody | Thermo Fisher Scientific | A21203 | 1/500 |
| Alexa Fluor® 488 donkey anti-rabbit IgG (H + L) | Thermo Fisher Scientific | A21206 | 1/500 |
| Alexa Fluor® 488 donkey anti-mouse IgG (H + L) | Thermo Fisher Scientific | A21202 | 1/500 |

The table shows the commercially available antibodies which were used for immunohistochemical analyses.

## Quantification of blood vessel alignment

Anisotropic alignment of endothelial tubes and collagen fibers was assessed via two-dimensional fast Fourier transform (2D-FFT) analysis, as previously described (Aw et al, 2016). Briefly, uncompressed images of endothelial tubes labeled with anti-CD31 antibody and basement membranes labeled with anti-COL1 antibody were analyzed with the FFT function of ImageJ, and the radial summation of the FFT frequency plot was calculated via the Oval profile plug-in. The degree of fiber alignment was reflected by the shape and height of the major peak in the FFT alignment plot. Images with oriented blood vessels resulted in a prominent peak centered at the principal axis of fiber alignment, whereas images with unaligned actin fibers resulted in an alignment plot with a broad peak or no peak.

## Expression analysis of fibroblast marker proteins

The number of cells expressing fibroblast markers, as well as the corresponding expression areas, were quantified using binary image analysis with ImageJ Fiji. For each skin equivalent condition, nine randomly selected dermal regions measuring $1000 \times 500\,\mu m$ were subjected to analysis.

## Single-cell RNA-seq library preparation

NHEKs, NHDFs, and HUVECs cultured to a subconfluent state were harvested through trypsinization according to the product protocol. Briefly, the cells were washed with HEPES-buffered solution (Kurabo) to remove the medium and treated with 0.025% trypsin/EDTA solution (Kurabo) for 3–7 min, followed by the addition of a trypsin-neutralizing solution (Kurabo) and centrifugation to obtain a cell pellet. Constituent HSE cells were harvested through enzymatic digestion. On day 14 of air–liquid interface culture, the epidermis and dermis of the HSE were mechanically separated using forceps. The epidermal layer was enzymatically dissociated with 0.025% trypsin/EDTA solution, while the dermal layer was incubated in 10% FBS-supplemented DMEM containing 5000 U/ml collagenase type 1 (Worthington Biochemical Corp., NJ, USA) in a 37 °C water bath. The resulting cell suspension was filtered through a 70-μm cell strainer to remove debris and centrifuged to obtain a cell pellet. Single-cell RNA sequencing was conducted using the Chromium Next GEM

Single Cell 3′ Kit v3.1, Chromium Next GEM Chip G Single Cell Kit, and Dual Index Kit TT Set A (10x Genomics) according to the manufacturer's protocol (CG000315 Rev E). Cell viability and count were assessed using a Countess 3 Automated Cell Counter following trypan blue staining and visual inspection. The cell suspension was processed on a chromium controller for gel bead-in-emulsion (GEM) generation, where polyadenylated mRNA was reverse transcribed to cDNA and uniquely barcoded. The cDNA was subjected to amplification, quality assessment via electrophoresis, and library preparation involving fragmentation, end-repair, A-tailing, adapter ligation, and PCR amplification. Library quality was evaluated using Agilent's High Sensitivity D5000 ScreenTape system, and the DNA concentration was measured with a Qubit 1X dsDNA HS Assay Kit on a Qubit 4 Fluorometer. Libraries were sequenced on an Illumina NovaSeq 6000 with PE150, generating ~120 Gb of data per sample from ~800 million reads.

## Single-cell RNA-seq data processing: In vitro skin data

FASTQ reads of 10× scRNA sequencing data were processed with the hg38 reference genome via Cell Ranger (v7.1.0) (Hao et al, 2024). A total of 27,082 cells passed the quality control steps. The processed matrices of different batches were merged, and the following analyses were performed via SEURAT-1 (v5.0.3). Cells with 1000 to 8000 RNA features and mitochondrial RNA contents of less than 7% were selected. The selected expression matrix was normalized by the total number of UMIs per cell and was log-transformed.

The filtered dataset was split into original sample identities via the Split Object function, followed by data log normalization of UMI counts and identification and the identification of 2000 more variable genes per sample.

The integrated data were then used for standard cell clustering and visualization with Seurat, which uses the 2000 most variable genes of the integrated dataset as input. Next, we used the function FindIntegrationAnchors with default parameters and 30 canonical correlation analysis (CCA) dimensions to identify the integration anchors between our five datasets. These anchors were subsequently used for integration via the IntegrateData function, again with the first 30 CCA dimensions and default parameters.

The integrated data were then used for standard cell clustering and visualization with Seurat, which uses the 2000 most variable genes of the integrated dataset as input. First, the data were scaled via the ScaleData function, and principal component analysis (PCA) dimensions were calculated via the RunPCA function. Next, unsupervised clustering of the data was performed with the FindNeighbors and FindClusters functions. For the FindNeighbors function, we used the first 30 PCA dimensions to construct a shared nearest neighbor (SNN) graph for our dataset (Solé-Boldo et al, 2020). Then, we clustered the cells with the function FindClusters via a shared nearest neighbor (SNN) modularity optimization-based clustering algorithm with a resolution of 0.7. Finally, for visualization, we used the RunUMAP function with default parameters and 30 PCA dimensions.

## Single-cell RNA-seq data processing: In vivo skin data and data integration with in vitro skin data

In vivo skin single-cell analysis was carried out via methods described in previous studies (Solé-Boldo et al, 2020), (GSE130973

(https://www.ncbi.nlm.nih.gov/geo/query/acc.cgi?acc=GSE130973), Data ref: Solé-Boldo et al, 2020). After mapping cell types based on the expression of marker genes for each cell type, we further explored the correspondence between the clusters and the four fibroblast subpopulations mentioned in the original paper. This was accomplished by examining the expression of marker genes for papillary fibroblasts and reticular fibroblasts, as well as the top-enriched Gene Ontology (GO) terms for the four fibroblast clusters. Consistent with previous studies, Cluster 1 was classified as secretory-reticular, Cluster 2 as pro-inflammatory, Cluster 3 as secretory-papillary, and Cluster 9 as mesenchymal.

To integrate in vitro and in vivo data, cells annotated as vascular ECs, pericytes, keratinocytes, and fibroblasts were initially extracted from the in vivo dataset. The cells from each sample were filtered under specific conditions: for in vitro samples, cells with 1000 to −8000 RNA features and mitochondrial RNA contents of less than 7% were selected; for in vivo samples, the criteria were fewer than 7500 RNA features and mitochondrial RNA contents of less than 5%. Following this, normalization was performed via the SCTransform method. Using Seurat's default approach, the top 2000 genes with the most variability from each dataset were extracted. All the data were then combined to compute principal component analysis (PCA) dimensions, and batch effects between datasets were mitigated using the Harmony algorithm's RunHarmony function (parameters set to theta = 1, max.iter.harmony = 10). Unsupervised clustering of the data were subsequently performed via the FindNeighbors and FindClusters functions. Finally, for visualization purposes, the RunUMAP function was employed with default parameters and 30 PCA dimensions.

## Cell-to-cell communication analysis

To analyze the estimated intercellular interactions established by different cell types present in artificial skin, we utilized R CellChat3 (v1.6.1) (Jin et al, 2021). This tool can quantitatively infer and analyze intercellular communication networks from scRNA-seq data and contains ligand–receptor interaction databases (https://github.com/sqjin/CellChat). In our approach, pairwise comparisons across all cell clusters present in the artificial skin dataset were conducted, and the analysis was performed using the default parameters set by the tool.

## Trajectory analysis of fibroblast-to-pericyte differentiation

Pseudotime trajectory analysis was performed using Monocle3 (version 1.3.1) to infer the developmental progression of cell populations (Cao et al, 2019; Trapnell et al, 2014). The integrated Seurat object was converted to a cell_data_set object using the SeuratWrappers package. Dimensionality reduction was performed using UMAP, followed by cell clustering with a resolution parameter of $1 \times 10^{-3}$. A principal graph was constructed using the learn_graph function with 2000 center points to capture the trajectory structure while maintaining computational efficiency. The parameter use_partition = FALSE was applied to allow trajectory learning across the entire selected cell population. Following pseudotime inference, to focus on the fibroblast-to-pericyte differentiation process, we extracted cells along a specific

trajectory path in the principal graph. The path was defined from node Y_184 to node Y_528, identified based on biological relevance and cell type composition. Cells were selected based on their proximity to this trajectory path using Euclidean distance in UMAP space, with a distance threshold set at the 20th percentile of all cell-to-path distances. To ensure data quality, genes were pre-filtered based on detection rate, retaining only those expressed (log-normalized count >0) in at least 10% of cells within the selected population (McCarthy et al, 2017). For each gene passing quality filters, we modeled expression as a smooth function of pseudotime using locally estimated scatterplot smoothing (LOESS) regression[69] with a span parameter of 0.3 and a degree 2 polynomial. To quantify the extent of systematic expression variation captured by the smoothed model, we calculated a dynamics range score for each gene:

$$Dynamics\,Range = \frac{\max(\hat{y}) - \min(\hat{y})}{\max(y) - \min(y)}$$

where y represents observed log-normalized expression values and $\hat{y}$ represents LOESS-fitted values along pseudotime. Genes with dynamic range scores below 0.05 were classified as "static" and excluded from downstream analyses. The remaining genes, classified as "dynamic," were further analyzed using Spearman's rank correlation with pseudotime. The top 20 genes ranked by absolute correlation strength in each direction (upregulated: $\rho > 0$; down-regulated: $\rho < 0$) were identified as key fate-determining genes. Statistical significance of correlations was assessed using Spearman's rank correlation test with Bonferroni correction where appropriate. Gene expression patterns along pseudotime were visualized using scatter plots with LOESS-smoothed trend lines.

### Trajectory analysis of fibroblast differentiation

Trajectory analysis of the fibroblast-directed differentiation branch was performed using the same Monocle3-based analytical framework as described for the pericyte-directed trajectory, with the following modifications:

Principal graph construction: The learn_graph function was executed with 200 center points instead of 2000.

Dynamic gene filtering: Genes with dynamic range scores <0.1 were classified as "static" and excluded from downstream analyses. All other analytical steps, including pseudotime inference, LOESS-based modeling, and correlation-based ranking of dynamic genes, were performed identically to the pericyte trajectory analysis.

### Evaluation of the epidermal barrier function of HSEs via TEWL measurement

Trans-epidermal water loss (TEWL) was measured in the HSEs (N = 4) using a Tewameter TM HEX (Courage and Khazaka electronic, Cologne, Germany) according to the instruction manual. Prior to collection, the skin equivalent was incubated inside the biosafety cabinet for 15 min to allow equilibration to the ambient temperature and humidity of the room. The measurement was repeated three times, and the average and standard deviation of each well were calculated.

### Measurements of skin elasticity

Skin elasticity was determined via a noninvasive, in vivo suction skin elasticity meter, the Cutometer MPA 580 (Courage & Khazaka electronic) (Ezure et al, 2009). Briefly, with a 2-mm probe, a negative pressure of 400 mbar was applied to the HSEs for a period of 2 s, followed by 2 s of relaxation time, and the ratio of immediate retraction (Ur) and the ability of redeformation of the skin (Ua) to final distension (Uf) was analyzed (R2 = Ua/Uf, R7 = Ur/Uf).

### Nutrient-poor culture of HSEs and evaluation of their responsiveness to ascorbic acid

All HSEs were treated with nutrient-poor medium (DMEM supplemented with 1% FBS, 1% penicillin/streptomycin, 5 μg ml$^{-1}$ insulin, and 1 μM hydrocortisone) at the air–liquid interface on day 3 and maintained for 12 d. This NP medium is composed of reducing the FBS concentration in the HSE growth medium from 10 to 1%, and by removing both AA and bFGF. Then, 500-μM ascorbic acid was added to the HSEs cultured at the air–liquid interface on day 5 and maintained for 10 days.

### Statistics and reproducibility

Microsoft Excel (Microsoft, Redmond, WA, USA) and BellCurve for Excel were used for statistical analysis. All values are expressed as the means ± standard deviations (SDs). Analysis of the samples was performed at least in triplicate, and the results were averaged. Differences between groups were considered significant at $P < 0.05$. All the experiments were repeated at least two times.

## Data availability

scRNA-seq data from this study have been deposited into the Sequence Read Archive (SRA) database under the BioProject ID. PRJNA1191968. The source data of this paper are collected in the following database record: BioStudies ID. S-BSST2270. The source code of this paper are reposited in the following database record: Zenodo, https://doi.org/10.5281/zenodo.17551418, (https://zenodo.org/records/17551418).

The source data of this paper are collected in the following database record: biostudies:S-SCDT-10_1038-S44319-026-00757-w.

## Peer review information

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

## Acknowledgements

We thank laboratory members at the Institute of Advanced Biomedical Engineering and Science, TWIns, and Tokyo Women's Medical University. We also thank Mr. M. Hashimoto, Y. Hayashi, and all the members of Rohto Pharmaceutical Co., Ltd., for their support and encouragement in this project. We would like to thank Rohto Pharmaceutical Co., Ltd., for funding support.

## Author contributions

**Shun Kimura**: Conceptualization; Data curation; Formal analysis; Funding acquisition; Validation; Investigation; Visualization; Methodology; Writing—original draft; Project administration. **Sachiko Sekiya**: Conceptualization; Supervision; Investigation; Methodology; Project administration; Writing—review and editing. **Sawa Yamashiro**: Software; Formal analysis; Investigation; Writing—original draft. **Tetsutaro Kikuchi**: Methodology; Writing—review and editing. **Masatoshi Haga**: Software; Supervision; Writing—review and editing. **Tatsuya Shimizu**: Conceptualization; Supervision; Project administration; Writing—review and editing.

Source data underlying figure panels in this paper may have individual authorship assigned. Where available, figure panel/source data authorship is listed in the following database record: biostudies:S-SCDT-10_1038-S44319-026-00757-w.

## Disclosure and competing interests statement

The authors declare competing financial interests related to the publication of this study, including direct investments in Rohto Pharmaceutical Co., Ltd.

# Expanded View Figures

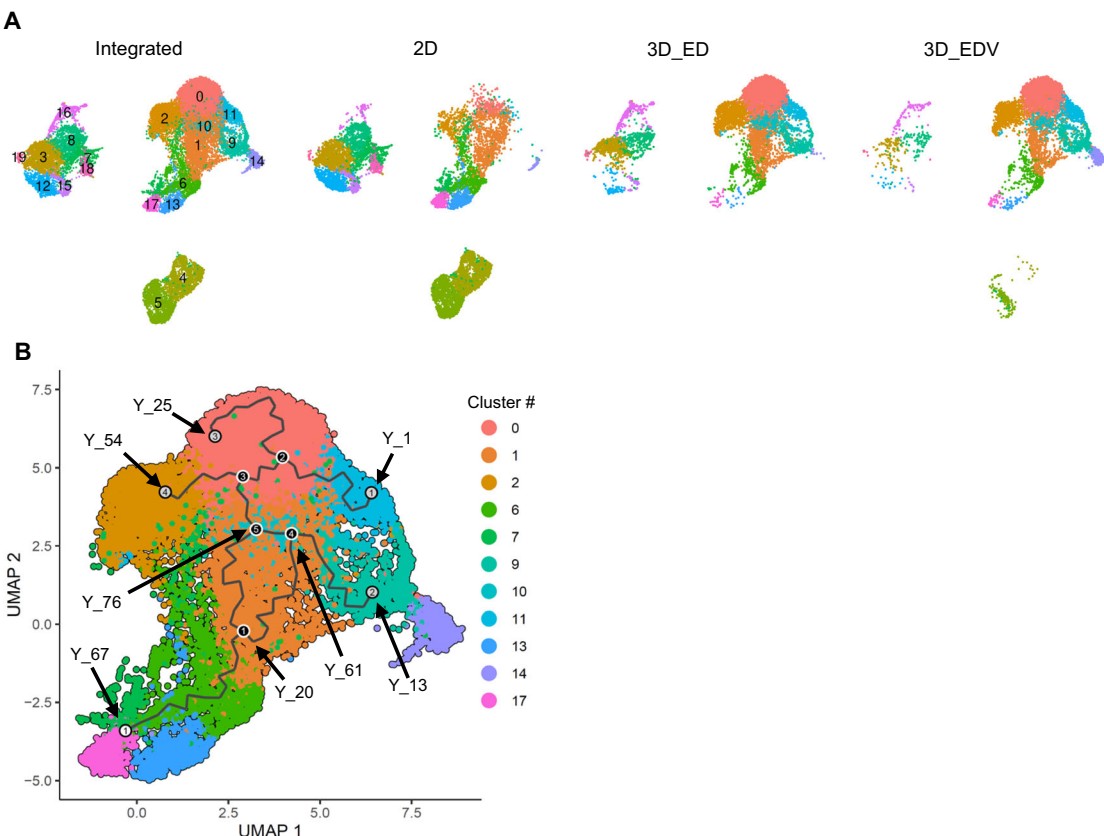

**Figure EV1.  scRNA-seq analysis of integrated in vitro samples.**

(**A**) UMAP plot of scRNA-seq for in vitro human skin data, and distribution of all cells under each of the three conditions (2D, 3D_ED, and 3D_EDV). (**B**)Visualization of the pseudotime trajectories of fibroblasts clusters distinguished on the UMAP plot.

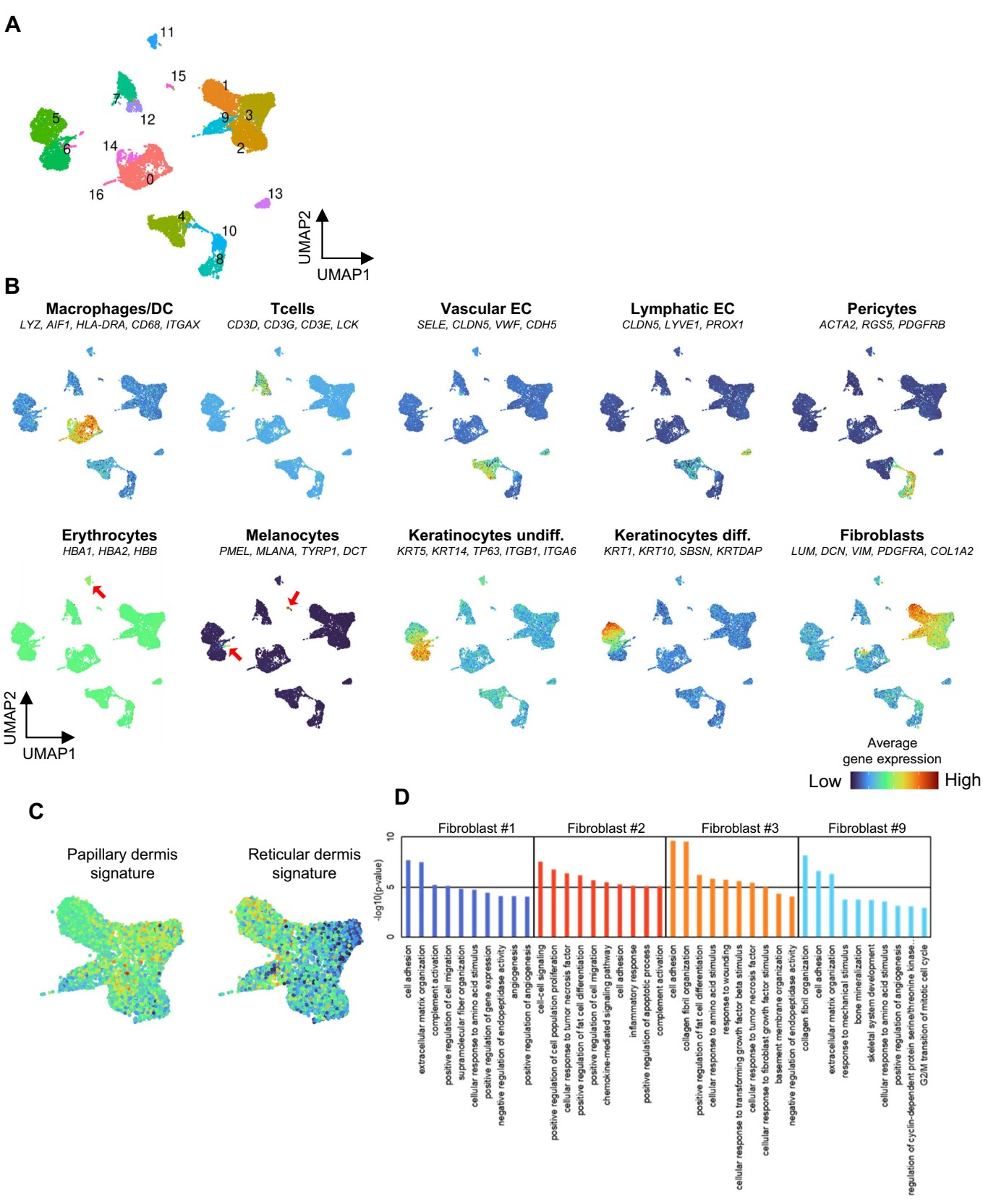

◀    **Figure EV2.   Re-evaluation of in vivo samples through scRNA-seq.**

In vivo skin single-cell analysis was carried out via methods described in previous studies(Solé-Boldo et al, 2020), (GSE130973, Data ref: Solé-Boldo et al, 2020). **(A)** A UMAP plot presenting single-cell transcriptomic data from whole human skin samples ($N = 5$). Each point signifies a single cell, with coloration based on unsupervised clustering executed using Seurat. **(B)** Mean expression of genes forming the papillary and reticular gene signatures applied to predict the dermal localization of fibroblasts within the four clusters. **(C)** A UMAP plot featuring the average expression of previously established cell type markers for distinguishing cell populations. Red denotes maximum gene expression, whereas blue represents minimal or non-existent expression of a specific gene set in log-normalized UMI counts. **(D)** The ten top significantly enriched GO terms within each fibroblast subpopulation, arranged by *P* value (hypergeometric test with Benjamini–Hochberg correction).

                                                                                      

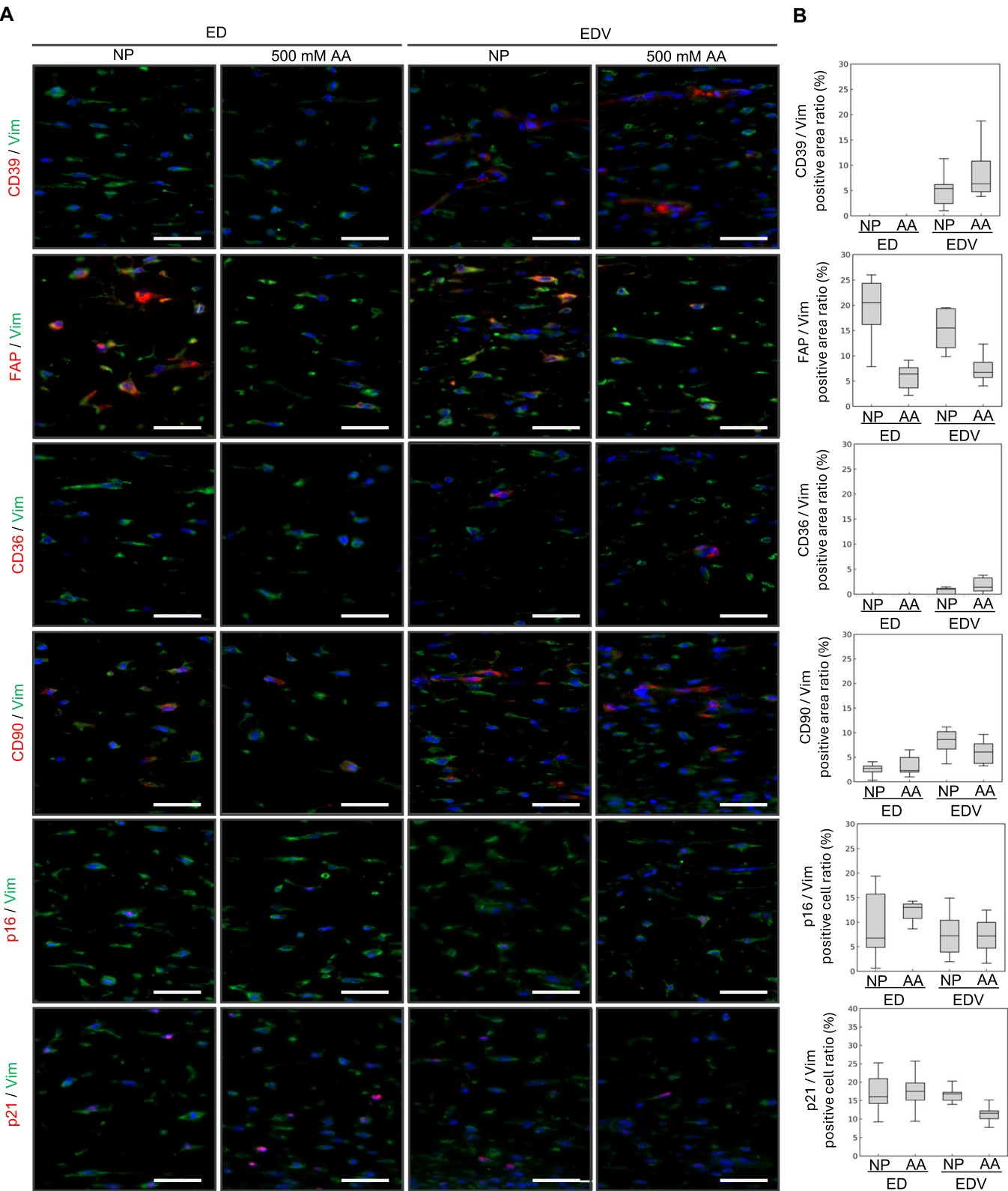

◀︎  **Figure EV3.  Immunohistological analysis of dermal mesenchymal cell distribution in the HSEs cultured under NP conditions.**

(**A**) Immunohistochemical analyses of HSEs cultured under NP conditions with or without AA. Scale bar, 50 μm. (**B**) Quantitative analysis of the ratio of fibroblast marker-positive area. Box plots depict the median as the center line, the interquartile range as the box, and whiskers extending to 1.5 × the interquartile range. Outliers were omitted. $N = 9$, technical replicates.

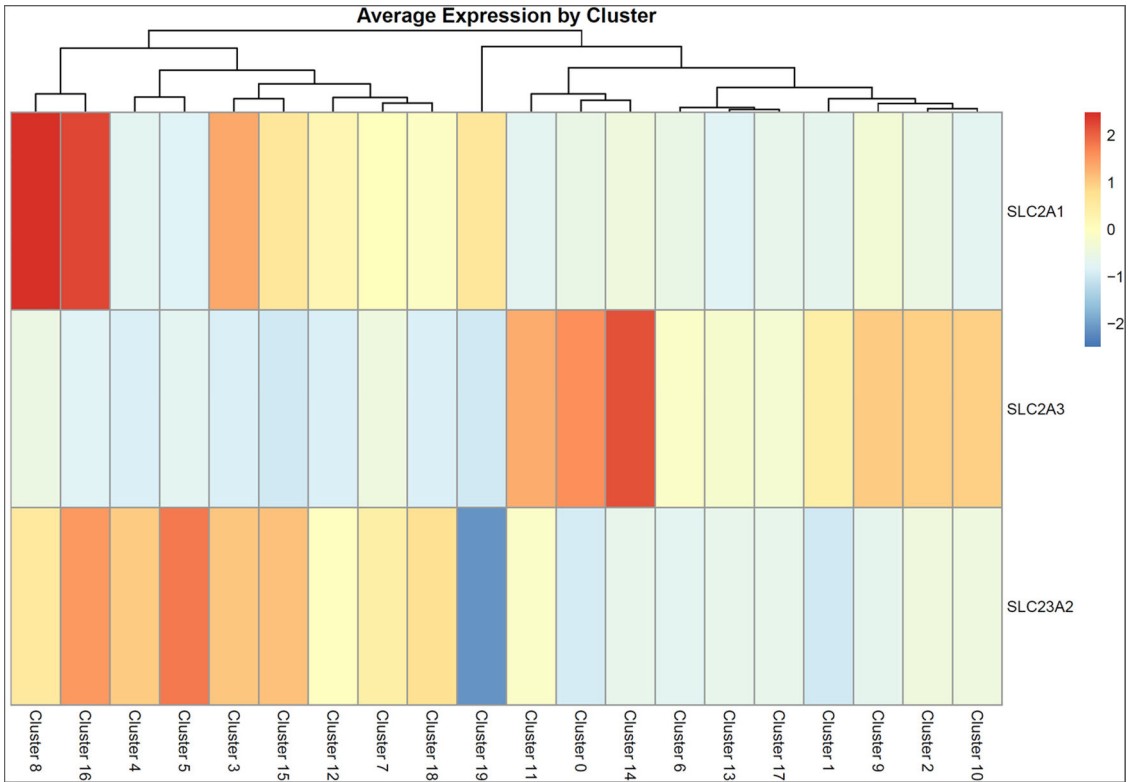

**Figure EV4. Cluster-specific mean expression patterns of ascorbic acid and dehydroascorbic acid transporter genes.**

Expression levels of ascorbic acid and dehydroascorbic acid transporter genes with a maximum mean expression above 0.01 are shown. Expression levels were scaled to Z-scores across clusters and visualized as a heatmap.

