## [Peer Review File · EMBO Reports]

The heterogeneity of dermal mesenchymal cells reproduced in skin equivalents regulates barrier function and elasticity

Shun Kimura, Sachiko Sekiya, Sawa Yamashiro, Tetsutaro Kikuchi, Masatoshi Haga, and Tatsuya Shimizu

Corresponding author(s): Shun Kimura (skimura@rohto.co.jp)

Review Timeline:

Transfer Date:	21st Jul 25
Editorial Decision:	5th Aug 25
Revision Received:	11th Nov 25
Editorial Decision:	8th Dec 25
Revision Received:	23rd Dec 25
Editorial Decision:	26th Jan 26
Revision Received:	6th Feb 26
Accepted:	9th Mar 26

Editor: Deniz Senyilmaz Tiebe / Martina Rembold

Transaction Report: This manuscript was transferred to EMBO reports following peer review at Review Commons.

**Review
COMMONS**

Review #1

1. Evidence, reproducibility and clarity:

Evidence, reproducibility and clarity (Required)

The manuscript by Kimura et al investigates the role of different cell populations in the development of human skin equivalents (HSEs). The observe that the addition of vascular endothelial cells to HSEs improves epidermal differentiation and barrier function, alongside differentiation of fibroblasts into papillary, reticular, and pericyte like mesenchymal cells. The authors also use single-cell transcriptomics to characterise the gene signatures and putative signalling pathway in the fibroblasts. Finally, the authors use nutrient poor medium and ascorbic acid to modulate HSE develop.

One of the most significant questions arising from the findings is how the presence of vasculature can induce differentiation of fibroblasts from a common population, especially given that previous studies have shown that fibroblast identity is programmed during development. Some specific comments and suggestions for improving the manuscript are listed below.

****Major points:****

1. The introduction describes the effects of different environmental cues and aging on fibroblast phenotype, but it would be good to note the developmental origins of dermal fibroblasts, which specifies their fate and function (Driskell et al, Nature 2013).
2. In Fig 2, how do TEWL measurements compare to constructs without an epidermal layer or human skin? It may seem obvious that barrier function would be negligible in these models, but it would be a helpful negative control for interpreting the relative effects of vasculature on barrier function.
3. The mechanical measurements in Fig 2 are a nice idea, but it is a bit difficult to interpret without comparison to other conditions (e.g. human skin) or by reporting more universal mechanical parameters (e.g. Young's modulus).
4. The induction of region-specific fibroblast markers is interesting and a bit unexpected since all the fibroblasts came from the same source before seeding into HSEs. The conclusions require additional support from quantification of the IF staining in Fig 3.
5. Likewise, could the authors clarify whether the cells were passaged before seeding into the HSE, and if so, what passage number. Could passaging affect the responses observed? Please add a discussion point about this.
6. The scRNA-seq suggests that the in vitro populations do not discriminate between

secretory papillary and pro-inflammatory fibroblasts. Could the authors add some further analysis or discussion regarding this point?

7. In Fig 6, it will be important to add quantification of epidermal thickness and differentiation marker expression to support the conclusions.

8. A key question is how NP and AA conditions affect the fibroblast populations as this seems to be a key factor in HSE maturation and would then link back to the previous sections. It would be good to stain for fibroblast markers in these samples.

9. As noted above, the ability of the vasculature to direct differentiation of a common fibroblast population into different phenotypes is one of the key findings of the study. To strengthen these observations, could additional analysis of the transcriptional data be possible. For example, would trajectory analysis potentially show how the different populations are evolving or related? In addition, could the CellChat analysis be performed between the vasculature and the different populations in Fig 5, which are mapped to in vivo populations? This might be a more relevant analysis than the populations in Fig 4.

****Minor points:****

1. The abstract states that enabling in vitro evaluation of drug efficacy using methodologies that are identical to those used in human clinical studies. This seems to be an over interpretation of the study and not well supported by the data. Please consider revising or removing.

2. Check referencing formatting in lines 118-121

2. Significance:

Significance (Required)

Overall, the study represents a systematic analysis of how vasculature contributes to skin model development, and the impact on fibroblast differentiation is an interesting observation. It would have been more impactful if some of the pathways and genes were followed up with mechanistic studies, but the findings are still useful to the field. Likewise, further insight into exactly how the vasculature regulates fibroblast phenotype would add to the impact as this is an unexpected but important finding.

3. How much time do you estimate the authors will need to complete the suggested revisions:

Estimated time to Complete Revisions (Required)

(Decision Recommendation)

Between 1 and 3 months

4. Review Commons values the work of reviewers and encourages them to get credit for their work. Select 'Yes' below to register your reviewing activity at Web of Science Reviewer Recognition Service (formerly Publons); note that the content of your review will not be visible on Web of Science.

Yes

Review #2

1. Evidence, reproducibility and clarity:

Evidence, reproducibility and clarity (Required)

In this study, the authors present a novel and well-executed approach to reconstructing human skin equivalents (HSEs) that more faithfully replicate the functional complexity of native skin by incorporating the natural heterogeneity of dermal mesenchymal cells, including spatially organized pericytes, papillary fibroblasts, and reticular fibroblasts. Through autonomous interactions among keratinocytes, fibroblasts, and vascular endothelial cells, the fully tricellular EDV model emerged as the most functionally complete among seven engineered HSE variants, demonstrating enhanced epithelialization, barrier integrity, dermal elasticity, and angiogenic architecture. The study's strengths lie in its realistic aging induction via nutrient deprivation by mimicking aspects of vascular insufficiency in the papillary dermis, and its integration of diverse and rigorous evaluation methods, including histological and molecular analyses (Ki67, ECM markers), barrier function (TEWL), and mechanical testing. Notably, ascorbic acid treatment improved epidermal turnover and extracellular matrix organization, particularly through effects on perivascular niche cells, highlighting its translational relevance for anti-aging interventions. Although the EDV model showed superior elasticity via suction testing, more comprehensive mechanical characterization and longitudinal ECM analysis could further elucidate how mesenchymal heterogeneity supports biomechanical resilience. Overall, this work underscores the importance of multicellular crosstalk in skin physiology and positions the EDV model as a robust in vitro platform with high relevance for regenerative medicine, aging research, and therapeutic screening, offering the potential to eliminate animal models in skin biology.

****Major comments:****

Despite its strengths, the study has several limitations that warrant further investigation. The authors describe a "senescent-like" phenotype under nutrient-poor (NP) conditions, yet do not provide direct evidence of cellular senescence using canonical markers such as SA- β -gal staining, p16^{INK4a} or p21 expression, or SASP profiling-weakening their aging-related conclusions.

The 500 μ M dose of ascorbic acid (AA), while within the reported range for skin models, is at the higher end compared to commonly used concentrations (100-300 μ M) and lacks justification via dose/response data. Normal physiological levels and changes in aging dermis should be referenced in discussion. AA is also an additive in their standard HSE media, but this was not sufficiently emphasized to draw attention. Would its removal from the baseline media make a difference?

Mechanistically, fibroblast heterogeneity is attributed to keratinocyte and vascular signals, but the signaling pathways involved (e.g., Wnt, TGF- β , VEGF) are not directly examined. Validating which paracrine factors (VEGF, PDGF, LAMA5, KGF) are mediating fibroblast transitions using inhibitors or RNA profiling could shed more light.

****Minor comments:****

The role of pericytes is also underexplored; while their presence is confirmed, functional assays or transcriptomic analyses to elucidate their contribution to ECM remodeling or vascular stability are not fully explored.

The origin of pericyte-like cells remains uncertain without lineage tracing or barcoding to distinguish whether they derive from fibroblasts, endothelial cells, or culture artifacts. Since they observe induced differentiation of fibroblast-like cells in 3D culture, it would be compelling to reconstruct differentiation trajectories (pseudotime analysis) from progenitor states to papillary/reticular/pericyte-like states from their scRNAseq data. Although AA enhanced collagen production and elasticity in the vascularized EDV model, the lack of response in the ED model is not addressed mechanistically.

The omission of immune cells which are key players in skin aging and homeostasis could increase physiological relevance of the model.

The exclusive use of standard HUVECs may not fully capture the behavior of tissue-specific microvascular endothelial cells, potentially limiting the fidelity of the vascular niche.

2. Significance:

Significance (Required)

This study presents a robust and innovative approach to human skin equivalent (HSE) reconstruction by integrating pericyte-like and endothelial cells with dermal fibroblast subtypes, using only commercially available cell types. A key strength lies in its ability to recapitulate aspects of in vivo fibroblast heterogeneity, including papillary, reticular, and perivascular populations, and to demonstrate functional consequences on tissue architecture, barrier integrity, ECM dynamics, and mechanical properties under aging-like, nutrient-poor conditions. The spontaneous emergence of a pericyte-like population without relying on freshly isolated primary pericytes or complex sorting protocols represents a methodological advance that increases the model's accessibility and scalability. Furthermore, the use of ascorbic acid to reverse aging-associated features in a vascular cell-dependent manner adds a compelling functional dimension, linking cell composition with therapeutic response.

Compared to existing models that either lack vascular cell compartments or do not account for dermal fibroblast heterogeneity, this study fills an important gap at the intersection of skin aging, vascular biology, and mesenchymal-epithelial interactions. The advance is both conceptual by elucidating the role of vascular and perivascular cells in shaping fibroblast identity and function and methodological, through the generation of a human skin model that approximates in vivo complexity without requiring animal models or ethically limited human tissue. The work will be of strong interest to basic science researchers in dermatology, tissue engineering, and aging, and has potential influence in regenerative medicine, cosmetic science, and drug screening, especially in the context of skin repair and anti-aging therapies. The audience is broad but most relevant to specialized communities in skin biology, mesenchymal cell biology, vascular biology, and organoid modeling, and may also attract attention from those developing non-animal testing platforms in applied and translational settings.

As a reviewer with expertise in inflammatory skin disease modeling using both animal systems and 3D organoid cultures, I bring a critical understanding of how cellular composition, microenvironmental cues, and co-culture conditions influence skin physiology and pathology. My interest in developing advanced co-culture systems to recapitulate human skin complexity positions me well to evaluate the relevance, innovation, and translational potential of this vascularized HSE model. I am especially qualified to assess the biological fidelity of the reconstructed skin architecture, the functional outcomes of introducing pericyte-like populations, and the implications of nutrient deprivation and ascorbic acid supplementation as aging-relevant perturbations.

3. How much time do you estimate the authors will need to complete the suggested revisions:

Estimated time to Complete Revisions (Required)

(Decision Recommendation)

Between 3 and 6 months

Yes

Review #3

1. Evidence, reproducibility and clarity:

Evidence, reproducibility and clarity (Required)

The study develops a tricellular human skin equivalent (HSE) model incorporating epidermal keratinocytes (NHEKs), dermal fibroblasts (NHDFs), and vascular endothelial cells (HUVECs). This model autonomously organizes pericytes, papillary fibroblasts, and reticular fibroblasts, mimicking in vivo dermal mesenchymal heterogeneity. The EDV model (all three cell types) demonstrates enhanced epidermal barrier function (reduced TEWL), dermal elasticity, collagen deposition, and vascular organization compared to simpler models. Single-cell RNA-seq confirms the emergence of pericyte-like and fibroblast subpopulations resembling in vivo counterparts. Nutrient-poor (NP) culture replicates aging phenotypes (reduced proliferation, barrier dysfunction, disordered collagen), rescued by ascorbic acid (AA), highlighting vascular cells' role in skin homeostasis. However, several key methodological clarifications (e.g., heatmap normalization, statistical reporting), more precise qualification of certain claims, and enhanced contextualization within the literature are needed before the work can be considered suitable for publication; I therefore recommend major revision.

****Major comments:****

1. Are the key conclusions convincing?

The core claim-that tricellular interactions recapitulate dermal mesenchymal heterogeneity and enhance skin functionality-is well-supported by histology, immunohistochemistry, functional assays (TEWL, elasticity), and scRNA-seq.

2. Should the authors qualify some of their claims as preliminary or speculative, or remove them altogether?

The assertion that HSEs enable "identical" methodology to clinical studies (p. 2, line 29) is exaggerated. While elasticity was measured via Cutometer (used clinically), the model lacks immune/neural components and long-term stability for full translational equivalence.

3. Would additional experiments be essential to support the claims of the paper? Request additional experiments only where necessary for the paper as it is, and do not ask authors to open new lines of experimentation.

Adequacy of Experimental Evidence & Need for Additional Experiments: No essential control appears to be missing: the authors include conditions {plus minus}ascorbic acid and {plus minus}vascular cells to isolate those effects. One could suggest a few additional experiments to further bolster the conclusions, but they are not strictly required for the main message. For example, to pinpoint the contribution of each mesenchymal subset, the authors could engineer HSE variants lacking one component at a time (omit pericytes or use only papillary vs. only reticular fibroblasts) to see how each omission affects barrier or elasticity. This would directly confirm each cell type's role. However, such experiments may be technically involved (especially isolating pure papillary vs. reticular fibroblast populations and ensuring viability in 3D culture) and might be beyond the scope of a single study. Another possible extension could be mechanistic assays, such as examining specific molecular signals: e.g., testing if blocking known paracrine factors from pericytes or fibroblast subsets diminishes the observed improvements. Given that pericytes can secrete laminin-511 and other factors that promote keratinocyte growth, the authors might, in future work, explore whether such factors mediate the enhanced epidermal proliferation seen with the vascularized HSE. Overall, the current data are sufficiently convincing that additional experiments are not absolutely necessary for publication.

4. Are the suggested experiments realistic in terms of time and resources? It would help if you could add an estimated cost and time investment for substantial experiments

-

5. Are the data and the methods presented in such a way that they can be reproduced?

Yes

6. Are the experiments adequately replicated and statistical analysis adequate?

The manuscript's data are presented in a manner that generally supports reproducibility.

The authors state that all data are presented as "mean {plus minus} SD" (Methods, p.36).

This is acceptable and clearly reported. However, I suggest that the authors consider using mean {plus minus} SEM for specific datasets where the primary goal is to assess statistical significance between groups - for example, for the Ki67-positive cell proliferation data (Fig. 6c) - as SEM better reflects the precision of the group mean for inferential comparisons. In contrast, for functional measures that inherently exhibit biological variation across samples (e.g., TEWL, skin elasticity), using mean {plus minus} SD remains fully appropriate, as SD reflects true inter-sample variability. To improve clarity and reproducibility, I encourage the authors to briefly state in the Methods or figure legends why SD or SEM is used in each case, in line with best practice guidelines.

****Minor comments:****

1. For Figure 4e, it would be helpful if the authors could clarify in the figure legend or Methods whether the heatmap shows log-normalized expression values (as derived from the Seurat object) or z-scored expression across cells or samples. This distinction affects the interpretation of relative versus absolute expression levels of the collagen and elastic fiber-related genes, which are central to the study's conclusions about ECM remodeling.
2. Typos: "factr" → "factor" (p. 16, line 244); "severl" → "several" (p. 22, line 367).

2. Significance:

Significance (Required)

The study innovatively reconstructs dermal mesenchymal heterogeneity using commercially available cells and autonomous tricellular interactions, bypassing costly cell-sorting approaches. This democratizes complex HSE models for broader labs. This study demonstrates that vascularization is critical not only for nutrient supply but for instructing fibroblast/pericyte differentiation and ECM organization. The NP+AA paradigm (Fig. 6) offers a facile in vitro model for skin aging interventions, highlighting AA's efficacy via perivascular mechanisms.

Audience: Tissue engineers, dermatologists, cosmetic/pharma researchers (anti-aging screening), and developmental biologists studying mesenchymal niche regulation.

Placement in existing literature: Recent advances in skin tissue engineering have highlighted the importance of dermal fibroblast heterogeneity in skin homeostasis and regeneration. Single-cell transcriptomic studies (Tabib et al., J Invest Dermatol 2018; Solé-

Boldo et al., Commun Biol 2020) have established that papillary and reticular fibroblasts exhibit distinct gene expression and functional roles. Prior engineered skin models incorporating fibroblast subtypes (Moreira et al., Biomater Sci 2023) or pericytes (Paquet-Fifield et al., J Clin Invest 2009) demonstrated improvements in vascularization or epidermal differentiation. However, a unified 3D human skin equivalent integrating vascular cells, pericytes, and spatially organized fibroblast subpopulations has not been systematically achieved. The present work by Kimura et al. advances the field by demonstrating that autonomous interaction among keratinocytes, endothelial cells, pericytes, and heterogeneous fibroblasts significantly enhances both barrier function and dermal elasticity, thus bringing engineered skin models closer to physiological skin. This addresses a key gap between prior single-cell descriptive studies and functional tissue engineering.

Define your field of expertise with a few keywords: experimental dermatology, skin cancer, tissue engineering and 3D skin models, cell biology, tumor microenvironment, and the skin microbiome and barrier function.

3. How much time do you estimate the authors will need to complete the suggested revisions:

Estimated time to Complete Revisions (Required)

(Decision Recommendation)

Less than 1 month

4. Review Commons values the work of reviewers and encourages them to get credit for their work. Select 'Yes' below to register your reviewing activity at Web of Science Reviewer Recognition Service (formerly Publons); note that the content of your review will not be visible on Web of Science.

No

Review #4

1. Evidence, reproducibility and clarity:

Evidence, reproducibility and clarity (Required)

The manuscript by Kimura et al. define how epidermal morphogenesis in human skin equivalents (HSE) differ by combining vascular endothelial cells, epidermal keratinocytes, and dermal fibroblasts using staining and single-cell RNA-sequencing (scRNA-seq). The three cell system (EDV) displayed higher levels of Ki67+ cells, decreased levels of TEWL, and higher elasticity in comparison to the keratinocyte and fibroblast HSE system (ED). The overall structural morphology between the two systems is quite similar, though the expression of cytokeratin markers varies. EDV organoids specifically express COL1 and COL4 collagen markers surrounding the blood vessels. VEGF-VEGFR1 signaling between endothelia-fibroblasts seems to be pronounced in the EDV organoids according to scRNA-seq, suggesting active signaling between these two cell types. And ascorbic acid appeared to help nutrient poor ED and EDV organoids proliferate compared to controls. This work is well detailed and interesting, helping to define how endothelial cells function to make HSE organoids more faithfully mimic in vivo human skin. Only minor clarifications detailed below are needed.

1. The human skin control in Fig. 1c seems thinner than normal and would suggest that the ED and EDV models are hyperproliferative. Replacing the control with one that shows normal thickness would prevent incorrect conclusions of the data.
2. KI67 and TEWL readings for human skin as controls for Fig. 2b-c would help gauge how the organoids perform and whether they are abnormal. What is the elasticity index for facial sagging?
3. Ascorbic acid utilizes SLC23A1 and SLC23A2 to transport across cell membranes. Are their expression more pronounced in cluster 14 fibroblasts? This would help connect the scRNA-seq data to the ascorbic acid experiments.
4. There seems to be quite a bit of variability between replicant immunostains, in particular, vimentin in Fig. 3. Can the authors discuss this variability and whether any of the HSE organoid combinations reduced this variability?
5. Please provide number of replicates throughout figure legends.
6. Line 148 states "E and EV models were transparent and extremely soft", should read "E and ED models".
7. Line 150-151 states "In the E and EV models, an abnormal epidermis lacking a basal cell layer formed". The Krt5 staining in Figure 2 clearly shows a basal cell layer in these models, albeit abnormal. Stating that this the abnormal epidermis displayed a disrupted basal cell layer or columnar shape of basal cells were disrupted is more appropriate. In addition, these results do not show "crosstalk between NHEKs and NHDFs is essential for epithelialization" as the E and EV organoid models show epithelial stratification.

2. Significance:

Significance (Required)

This work is well detailed and interesting, helping to define how endothelial cells function to make HSE organoids more faithfully mimic in vivo human skin. Only minor clarifications detailed below are needed.

3. How much time do you estimate the authors will need to complete the suggested revisions:**Estimated time to Complete Revisions (Required)****(Decision Recommendation)**

Less than 1 month

No

Dear Dr. Kimura,

Thank you for submitting your manuscript to EMBO Reports, which was previously reviewed at Review Commons.

Referees express interest in your study presenting a human skin equivalent model, which explores the role of vascular endothelial cells in the spatial organization of the cellular components. However, they also raise concerns that need to be addressed to consider publication in EMBO Reports.

Having looked at all documents, we would like to invite you to submit a revised manuscript as in your revision plan. Please revise your manuscript with the understanding that the referee concerns (as in their reports) must be fully addressed and their suggestions taken on board. Please address all referee concerns in a complete point-by-point response. Acceptance of the manuscript will depend on a positive outcome of a second round of review. It is EMBO reports policy to allow a single round of major experimental revision only and acceptance or rejection of the manuscript will therefore depend on the completeness of your responses included in the next, final version of the manuscript.

We realize that it is difficult to revise to a specific deadline. In the interest of protecting the conceptual advance provided by the work, we recommend a revision within 3 months. Please discuss the revision progress ahead of this time with me if you require more time to complete the revisions, or if you have questions or comments regarding the revision (also by video chat).

1. A data availability section providing access to data deposited in public databases is missing (where applicable).
2. Your manuscript contains statistics and error bars based on $n=2$. Please use scatter plots in these cases.

You can submit the revision either as a Scientific Report or as a Research Article. For Scientific Reports, the revised manuscript can contain up to 5 main figures and 5 Expanded View figures, and it should not exceed 27000 characters. If the revision leads to a manuscript with more than 5 main figures it will be published as a Research Article. In this case the Results and Discussion section should be separate. If a Scientific Report is submitted, these sections have to be combined. This will help to shorten the manuscript text by eliminating some redundancy that is inevitable when discussing the same experiments twice. In either case, all materials and methods should be included in the main manuscript file.

4) a .docx formatted letter INCLUDING the reviewers' reports and your detailed point-by-point responses to their comments. As part of the EMBO publication's Transparent Editorial Process, EMBO reports publishes online a Review Process File (RPF) to accompany accepted manuscripts. This File will be published in conjunction with your paper and will include the referee reports, your point-by-point response and all pertinent correspondence relating to the manuscript.

<https://www.embopress.org/page/journal/14693178/authorguide#transparentprocess>

5) a complete author checklist, which you can download from our author guidelines <https://www.embopress.org/page/journal/14693178/authorguide>. Please insert information in the checklist that is also reflected in the manuscript. The completed author checklist will also be part of the RPF.

6) Please note that all corresponding authors are required to supply an ORCID ID for their name upon submission of a revised manuscript (. Please find instructions on how to link your ORCID ID to your account in our manuscript tracking system in our Author guidelines

Additional information on source data and instruction on how to label the files are available:

<https://www.embopress.org/page/journal/14693178/authorguide#sourcedata>

9) Our journal encourages inclusion of *data citations in the reference list* to directly cite datasets that were re-used and obtained from public databases. Data citations in the article text are distinct from normal bibliographical citations and should directly link to the database records from which the data can be accessed. In the main text, data citations are formatted as follows: "Data ref: Smith et al, 2001" or "Data ref: NCBI Sequence Read Archive PRJNA342805, 2017". In the Reference list, data citations must be labeled with "[DATASET]". A data reference must provide the database name, accession number/identifiers and a resolvable link to the landing page from which the data can be accessed at the end of the reference. Further instructions are available at <http://www.embopress.org/page/journal/14693178/authorguide#referencesformat>

10) Regarding data quantification (see Figure Legends:

<https://www.embopress.org/page/journal/14693178/authorguide#figureformat>)

12) Please also note our reference format:

13) All Materials and Methods need to be described in the main text using our 'Structured Methods' format, which is required for

all research articles. According to this format, the Methods section includes a Reagents and Tools Table (listing key reagents, experimental models, software and relevant equipment and including their sources and relevant identifiers) followed by a Methods and Protocols section describing the methods using a step-by-step protocol format. The aim is to facilitate adoption of the methodologies across labs. More information on how to adhere to this format as well as a downloadable template (.docx) for the Reagents and Tools Table can be found in our author guidelines:
<https://www.embopress.org/page/journal/14693178/authorguide#structuredmethods>.

An example of a Method paper with Structured Methods can be found here:
<https://www.embopress.org/doi/10.15252/msb.20178071>.

I look forward to seeing a revised version of your manuscript when it is ready. Please let me know if you have questions or comments regarding the revision.

Kind regards,

Deniz Senyilmaz Tiebe

Deniz Senyilmaz Tiebe, PhD
Senior Scientific Editor
EMBO Reports

Response to Reviewers

General Statements

We would like to express our sincere gratitude to the reviewers for their time and effort in reviewing our manuscript. Their insightful comments and suggestions have greatly contributed to the improvement of our work. To address the reviewer's comments, we have performed additional experiments and computational analyses. Firstly, we conducted pseudotime trajectory analysis using Monocle3 and intercellular communication analysis using CellChat to elucidate the differentiation pathways and molecular mechanisms of fibroblast and pericyte subpopulations. Second, we analyzed the expression of fibroblast and cellular senescence markers in HSEs cultured under nutrient-poor (NP) conditions to explore the mechanisms underlying the enhanced responsiveness to ascorbic acid observed in the EDV model.

These analyses have deepened our understanding of the molecular mechanisms that govern the formation and functional diversity of dermal mesenchymal cell subtypes within the HSE model. While we fully agree on the importance of performing detailed mechanistic analyses, such as those involving specific inhibitors or cell ablation experiments, we regret that, due to the considerable time and cost required for HSE culture and analysis, we were unable to include such experiments in the present revision.

We have made sincere efforts to address all other concerns raised by the reviewers.

Response to Reviewer #1

We have studied your comments carefully and found that you understood the value and significance of our study in this field. We are grateful for your evaluation and valuable suggestions for our manuscript. Our specific responses are listed below:

Major points

1. The introduction describes the effects of different environmental cues and aging on fibroblast phenotype, but it would be good to note the developmental origins of dermal fibroblasts, which specifies their fate and function (Driskell et al, Nature 2013).

Our response:

In accordance with the reviewers' suggestions, we have incorporated a summary of prior research regarding the developmental origins of dermal fibroblasts into lines 56–60 of the Introduction.

2. In Fig 2, how do TEWL measurements compare to constructs without an epidermal layer or human skin? It may seem obvious that barrier function would be negligible in these models, but it would be a helpful negative control for interpreting the relative effects of vasculature on barrier function.

We appreciate your valuable comments regarding the accurate interpretation of TEWL measurements. Estimated TEWL values for human skin have been reported in a systematic review and meta-analysis by Kottner et al. Specifically, the estimated TEWL (95% CI) for individuals aged 18–64 years varies by anatomical site: 15.4 (13.9–17.0) g/m²h for the right cheek, 6.5 (6.2–6.8) g/m²h for the

Response to Reviewers

midvolar right forearm, and 36.3 (29.5–43.1) g/m²h for the right palm. In comparison, the TEWL of our EDV model was 9.68 g/m²h, a value relatively close to that of human skin.

We also considered measuring TEWL in artificial skin models lacking epidermis. However, we found that such models remain moist due to culture medium, and pressing the measurement probe against them risks water droplets adhering to the sensor and causing damage. Although we recognize the significance of this measurement as a negative control, we refrained from conducting it due to the limitations of the equipment.

This information has been added to the Results section, lines 189–194.

3. The mechanical measurements in Fig 2 are a nice idea, but it is a bit difficult to interpret without comparison to other conditions (e.g. human skin) or by reporting more universal mechanical parameters (e.g. Young's modulus).

We greatly appreciate your insightful comments regarding the interpretation of skin viscoelasticity measurements using the Cutometer. The Cutometer is a device that applies negative pressure to the skin to elevate its surface, allowing for the calculation of biomechanical properties based on the temporal changes in skin displacement. Notably, the R7 parameter—defined as the ratio of immediate retraction after pressure release to the maximum deformation during suction—has been shown to correlate significantly with age.

In this study, we evaluated HSEs under the same measurement conditions as those used in previous human clinical studies. Accordingly, we have cited past Cutometer data for human skin and discussed the relationship between those findings and our HSE measurements. These revisions have been made to lines 214–227.

We determined that performing Cutometer measurements on human skin would be impractical due to the ethical committee procedures and associated costs. Although evaluating Young's modulus using techniques such as AFM to assess the mechanical properties of collagen fibers is a fascinating and informative approach, we have opted not to pursue this analysis due to the substantial time and cost required for sample preparation.

4. The induction of region-specific fibroblast markers is interesting and a bit unexpected since all the fibroblasts came from the same source before seeding into HSEs. The conclusions require additional support from quantification of the IF staining in Fig 3.

Our response:

Thank you for your valuable advice on strengthening the conclusion of our manuscript. We conducted a quantitative analysis of regions positive for fibroblast and pericyte marker proteins within nine randomly selected areas measuring 1000×500 μm. The analysis revealed that the papillary fibroblast markers CD39 and FAP, as well as the pericyte markers NG2 and αSMA, were significantly more abundant in the EDV model compared to other models. Furthermore, regarding reticular fibroblast markers, CD36 was expressed in the DV and EDV models, while CD90 was prevalent in the ED and EDV models (Fig. 3b and 3d). These observations are reflected in lines 244–246 and lines 250–253.

Response to Reviewers

5. Likewise, could the authors clarify whether the cells were passaged before seeding into the HSE, and if so, what passage number. Could passaging affect the responses observed? Please add a discussion point about this.

Our response:

For all cell types, passage 4 or 5 cells were utilized for the reconstitution of human skin equivalents (HSE). Indeed, Philippeos et al. demonstrated that while CD39, CD90, and CD36 are detectable in primary CD31⁻CD45⁻Ecad⁺ dermal cells, the expression of CD39 is lost after a single passage. In contrast, CD90 and CD36 remain detectable for up to four passages. These findings underscore the impact of in vitro culture on the depletion of fibroblast marker expression. Since we employed NHDFs that had undergone four to five passages for HSE reconstruction, it is reasonable to assume that these cells had already lost specific fibroblast subpopulations, including CD39⁺ cells. Consistent with this, our scRNA-seq analysis revealed that most fibroblasts cultured in 2D formed an artificial population comprising cells in the S and G2M phases, along with secretory-reticular fibroblasts (Fig. 5c and 5d). Additionally, immunohistochemical analysis confirmed a near-complete absence of CD39⁺, CD90⁺, FAP⁺, NG2⁺, and αSMA⁺ cells in the dermis of both D and DV models, further indicating that serial passaging significantly reduces the expression of markers associated with papillary fibroblasts, reticular fibroblasts, and pericytes (Fig. 3). Interestingly, the introduction of vascular endothelial cells into the HSE appears to facilitate a partial restoration of fibroblast heterogeneity in cells passaged four to five times. However, whether this effect can be replicated in more extensively passaged fibroblasts remains to be verified. It is well established that excessive passaging induces cellular senescence, leading to reduced proliferative and differentiation capacities in mesenchymal stem cells. Therefore, it is conceivable that fibroblasts beyond a certain passage number may fail to recapitulate dermal mesenchymal cell heterogeneity, even in the presence of endothelial cells.

We have added this discussion to the revised manuscript on lines 435-449 and 454-461. However, due to the prolonged culture period required, we regret that we are unable to perform the additional validation experiments at this time.

6. The scRNA-seq suggests that the in vitro populations do not discriminate between secretory papillary and pro-inflammatory fibroblasts. Could the authors add some further analysis or discussion regarding this point?

Our response:

RNA-seq analysis of the *in vivo* samples reported by Boldo *et al.* demonstrated that papillary fibroblast markers were distributed across Fibroblast clusters #2 and #3, suggesting that papillary fibroblasts share partially overlapping gene expression patterns with pro-inflammatory fibroblasts (Fig. EV2c). Gene Ontology analysis further indicated that pro-inflammatory fibroblasts are functionally associated with intercellular communication involving immune and adipose cells (Fig. EV2d). However, such cell types are absent in our HSE model. This absence may have prevented pro-inflammatory fibroblasts from fully establishing their molecular identity, thereby resulting in their indistinguishability from papillary fibroblasts. This interpretation has been incorporated into the revised manuscript (Lines 408-417).

Response to Reviewers

7. In Fig 6, it will be important to add quantification of epidermal thickness and differentiation marker expression to support the conclusions.

Our response:

Thank you for your valuable advice regarding quantitative analysis. We quantified epidermal thickness based on image analysis, and the statistical results have been added to the revised manuscript (Fig. 6d and Lines 359-361).

8. A key question is how NP and AA conditions affect the fibroblast populations as this seems to be a key factor in HSE maturation and would then link back to the previous sections. It would be good to stain for fibroblast markers in these samples.

Our response:

We are grateful for your insightful comments, which are crucial for a more precise understanding of the physiological relevance of the NP culture model. In response, we performed immunohistochemical analyses of fibroblast markers (CD39, FAP, CD36, and CD90) and pericyte markers (NG2 and α SMA), followed by quantitative assessment of the marker-positive areas. The addition of ascorbic acid to the NP condition resulted in a significant reduction in the FAP-positive area in both the ED and EDV models, whereas no notable changes were observed for the other markers. These results have been incorporated into the revised manuscript (Fig. EV3 and Lines 368–374).

9. As noted above, the ability of the vasculature to direct differentiation of a common fibroblast population into different phenotypes is one of the key findings of the study. To strengthen these observations, could additional analysis of the transcriptional data be possible. For example, would trajectory analysis potentially show how the different populations are evolving or related? In addition, could the CellChat analysis be performed between the vasculature and the different populations in Fig 5, which are mapped to in vivo populations? This might be a more relevant analysis than the populations in Fig 4.

Our response:

As pointed out by reviewers, we acknowledge that elucidating the process and underlying mechanisms by which fibroblasts, whose heterogeneity is compromised in 2D culture, re-differentiate into distinct dermal mesenchymal subtypes constitutes a critical additional analysis to strengthen our findings.

To test the hypothesis that fibroblast cluster formation and maintenance depend on interactions with endothelial cells or keratinocytes, we applied pseudotime trajectory analyses by monocle3 and CellChat analyses. (Jin et al, 2024). First, we visualized the differentiation trajectories of fibroblast clusters distinguished on the UMAP plot (Fig. 4b). The analysis revealed that fibroblast clusters #6, #13, and #17, which were predominant in the two-dimensional culture environment, sequentially differentiated into clusters #1 (Y67-Y20) and #10 (Y20-Y76 and Y20-Y61) (Fig. EV1b). From cluster #10 fibroblasts, further differentiation into clusters #0 (Y76-Y25), #2 (Y76-Y54), #11 (Y76-Y1), and #9 (Y61-Y13) was observed (Fig. EV1b). To elucidate the differentiation trajectory leading to cluster #14 pericytes, we reran the pseudotime trajectory analysis by increasing the number of center points used in the learn_graph function from 200 to 2000. This analysis indicated that cluster #9 fibroblasts, derived from clusters #10 and #11, further differentiated into cluster #14 pericytes (Y184-Y528) (Fig. 4d).

To characterize the differentiation of each fibroblast and pericyte cluster and infer the underlying molecular mechanisms, we calculated Spearman's rank correlation coefficients between gene expression and pseudotime, selecting genes with absolute correlation values greater than 0.7. Only a small number of

Response to Reviewers

genes were identified along fibroblast-to-fibroblast differentiation trajectories, whereas 39 genes—including KDR (VEGFR2), EGFL6, FGFR3, and TGM2, which are associated with pericyte differentiation and vascular development—were extracted along the differentiation path from cluster #9 fibroblasts to cluster #14 pericytes (Table EV1). The fibroblast differentiation trajectories appear to be more complex than the fibroblast-to-pericyte differentiation pathway, and elucidating their regulatory mechanisms will require more careful and detailed analyses.

CellChat analysis between endothelial cells in Clusters #4 and #5 and Cluster #14 fibroblasts (pericytes) identified *PGF* and *VEGFB-VEGFR1* as characteristic signaling pathways, whereas *VEGFR2* was not detected. These findings suggest a potential involvement of *VEGFR1* and *VEGFR2* in the endothelial cell mediated regulation of fibroblast-to-pericyte differentiation. Because these two receptors are known to cooperatively modulate gene expression in HUVECs (Apte, Rajendra S. et al. Cell, Volume 176, Issue 6, 1248 - 1264), further elucidation of the underlying molecular mechanisms will require validation through inhibitor assays and studies using genetically modified cell lines. However, given that the reconstruction and reanalysis of the HSEs require more than three months, we have decided not to include these experiments in the current revision and instead consider them as important subjects for future investigation. The corresponding results and discussion have been incorporated into the revised manuscript (Lines 272–297, 481-494).

Reviewer #1

Minor points

1. The abstract states that enabling in vitro evaluation of drug efficacy using methodologies that are identical to those used in human clinical studies. This seems to be an over interpretation of the study and not well supported by the data. Please consider revising or removing.

Our Response:

Upon thorough consideration, we have deleted the statements that may be regarded as exaggerated (line 26-28 and 214-216).

2. Check referencing formatting in lines 118-121

Our Response:

We appreciate your attention to the reference format error. The necessary revisions have been completed.

Response to Reviewers

Response to Reviewer #2

We have studied your comments carefully and found that you understood the value and significance of our study in this field. We are grateful for your evaluation and valuable suggestions for our manuscript. Our specific responses are listed below:

Major comments:

1. Despite its strengths, the study has several limitations that warrant further investigation. The authors describe a "senescent-like" phenotype under nutrient-poor (NP) conditions, yet do not provide direct evidence of cellular senescence using canonical markers such as SA- β -gal staining, p16^{INK4a} or p21 expression, or SASP profiling-weakening their aging-related conclusions.

Our Response

Thank you for your valuable advice, which has helped clarify the physiological phenomena modeled by the NP condition. We performed immunohistochemical analyses of p16 and p21 and quantitatively evaluated the number of marker-positive cells. Across all experimental conditions, approximately 10–20% of the cells were positive for these senescence markers. However, no significant differences in the number of p16- or p21-positive cells were observed between the NP and AA conditions in either the ED or EDV models. Although a trend toward reduced p21-positive cell numbers was noted in the AA condition of the EDV model compared with the NP condition, the difference did not reach statistical significance (Fig. EV3).

The NP condition was designed to model the reduced supply of plasma components associated with age-related capillary loss. The 10-day culture period was sufficient for epidermal and dermal remodeling, as well as for detecting changes in epidermal barrier function and dermal elasticity associated with aging. However, it may have been insufficient to induce measurable alterations in the expression of senescence markers.

Based on these observations, we revised the description from “NP condition induces a senescence-like phenotype” to “NP condition induces disruption of skin barrier function and dermal elasticity” to avoid potential misinterpretation of the NP condition as a senescence model. The revision has been incorporated into the manuscript (Lines 338–339, 374-376, and 387-392).

2. The 500 μ M dose of ascorbic acid (AA), while within the reported range for skin models, is at the higher end compared to commonly used concentrations (100-300 μ M) and lacks justification via dose/response data. Normal physiological levels and changes in aging dermis should be referenced in discussion. AA is also an additive in their standard HSE media, but this was not sufficiently emphasized to draw attention. Would its removal from the baseline media make a difference?

Our Response

We sincerely appreciate the important comment regarding the rationale behind the ascorbic acid concentration used in the culture medium. As Reviewer 3 rightly pointed out, concentrations around 100–300 μ M are commonly employed in general in vitro assays. In our artificial skin model, we opted for a concentration of 500 μ M AA in the growth medium based on two considerations: (1) the model contains a high cell density of approximately 4×10^6 cells immediately after reconstruction, which is expected to result in substantial AA consumption, and (2) AA is not sufficiently stable in culture medium. Given the relatively long medium exchange interval of 48–72 hours, we deemed it necessary to maintain a certain AA level throughout this period. While no rigorous dose–response validation has been conducted, we have confirmed that this concentration does not induce toxicity or abnormalities in skin morphogenesis.

Response to Reviewers

As part of the revision, we considered revisiting the basal medium formulation; however, due to the significant time and resource demands, we have decided to forgo further optimization at this stage.

As described on lines 347–351, the NP medium was formulated to evaluate the potential impact of age-related declines in plasma component transport. We apologize for any confusion regarding the relationship between the HSE growth medium and the NP medium. In response to the reviewer's suggestion, we have added clarifying explanations and cautionary notes regarding the composition and rationale of these two media in both the Results and Methods sections (line 347-351 and 762-764).

3. Mechanistically, fibroblast heterogeneity is attributed to keratinocyte and vascular signals, but the signaling pathways involved (e.g., Wnt, TGF- β , VEGF) are not directly examined. Validating which paracrine factors (VEGF, PDGF, LAMA5, KGF) are mediating fibroblast transitions using inhibitors or RNA profiling could shed more light.

Our response:

To test the hypothesis that fibroblast cluster formation and maintenance depend on interactions with endothelial cells or keratinocytes, we applied pseudotime trajectory analyses by monocle3 and CellChat analyses. (Jin et al, 2024). First, we visualized the differentiation trajectories of fibroblast clusters distinguished on the UMAP plot (Fig. 4b). The analysis revealed that fibroblast clusters #6, #13, and #17, which were predominant in the two-dimensional culture environment, sequentially differentiated into clusters #1 (Y67-Y20) and #10 (Y20-Y76 and Y20-Y61) (Fig. EV1b). From cluster #10 fibroblasts, further differentiation into clusters #0 (Y76-Y25), #2 (Y76-Y54), #11 (Y76-Y1), and #9 (Y61-Y13) was observed (Fig. EV1b). To elucidate the differentiation trajectory leading to cluster #14 pericytes, we re-ran the pseudotime trajectory analysis by increasing the number of center points used in the learn_graph function from 200 to 2000. This analysis indicated that cluster #9 fibroblasts, derived from clusters #10 and #11, further differentiated into cluster #14 pericytes (Y184-Y528) (Fig. 4d).

To characterize the differentiation of each fibroblast and pericyte cluster and infer the underlying molecular mechanisms, we calculated Spearman's rank correlation coefficients between gene expression and pseudotime, selecting genes with absolute correlation values greater than 0.7. Only a small number of genes were identified along fibroblast-to-fibroblast differentiation trajectories, whereas 39 genes—including KDR (VEGFR2), EGFL6, FGFR3, and TGM2, which are associated with pericyte differentiation and vascular development—were extracted along the differentiation path from cluster #9 fibroblasts to cluster #14 pericytes (Table EV1). The fibroblast differentiation trajectories appear to be more complex than the fibroblast-to-pericyte differentiation pathway, and elucidating their regulatory mechanisms will require more careful and detailed analyses.

CellChat analysis between endothelial cells in Clusters #4 and #5 and Cluster #14 fibroblasts (pericytes) identified *PGF* and *VEGFB-VEGFR1* as characteristic signaling pathways, whereas *VEGFR2* was not detected. These findings suggest a potential involvement of *VEGFR1* and *VEGFR2* in the endothelial cell mediated regulation of fibroblast-to-pericyte differentiation. Because these two receptors are known to cooperatively modulate gene expression in HUVECs (Apte, Rajendra S. et al. Cell, Volume 176, Issue 6, 1248 - 1264), further elucidation of the underlying molecular mechanisms will require validation through inhibitor assays and studies using genetically modified cell lines. However, given that the reconstruction and reanalysis of the HSEs require more than three months, we have decided not to include these experiments in the current revision and instead consider them as important subjects for future investigation. The corresponding results and discussion have been incorporated into the revised manuscript (Lines 272–297, 481-494).

Response to Reviewers

Minor comments:

1. The role of pericytes is also underexplored; while their presence is confirmed, functional assays or transcriptomic analyses to elucidate their contribution to ECM remodeling or vascular stability are not fully explored.

The origin of pericyte-like cells remains uncertain without lineage tracing or barcoding to distinguish whether they derive from fibroblasts, endothelial cells, or culture artifacts. Since they observe induced differentiation of fibroblast-like cells in 3D culture, it would be compelling to reconstruct differentiation trajectories (pseudotime analysis) from progenitor states to papillary/reticular/pericyte-like states from their scRNAseq data.

Our response:

Human dermal pericytes promote the development and stabilization of both the epidermis and vasculature through the deposition of basement membrane components such as type IV collagen and laminin (Paquet-Fifield S *et. al.* J Clin Invest. 2009 Sep;119(9):2795-806.). The histological changes observed in the EDV model, accompanied by the differentiation of Cluster #14 fibroblasts with pericyte-like characteristics, are consistent with the known functions of pericytes. To verify the functionality of these pericyte-like cells and elucidate the underlying mechanisms, differentiation-inhibition or ablation experiments targeting Cluster #14 cells would be informative. Accordingly, we analyzed the signaling pathways regulating their differentiation and, as described in our response to Major Comment 3, identified several candidate molecules including VEGF receptors. We fully agree that additional experiments using VEGF-signaling inhibitors would provide valuable mechanistic insight; however, given the time and resources required, we plan to address these studies in future work.

2. Although AA enhanced collagen production and elasticity in the vascularized EDV model, the lack of response in the ED model is not addressed mechanistically.

Our response

To investigate the mechanism underlying the enhanced responsiveness of the EDV model to ascorbic acid, we examined two hypotheses. The first hypothesis proposed that NP conditioning alters the subpopulation composition or characteristics of dermal mesenchymal cell heterogeneity. Immunohistochemical analyses of papillary and reticular fibroblast markers (CD39, FAP, CD36, and CD90) and senescence-associated markers (p16 and p21) in the ED and EDV models revealed no apparent loss or substantial alteration of marker expression following NP conditioning (Fig. EV3). The second hypothesis posited that the dynamics of ascorbic acid uptake differ between the ED and EDV models. To test this, we compared the expression of transporter genes for ascorbic acid and dehydroascorbic acid among dermal mesenchymal cell subpopulations. Cluster #14 fibroblasts (pericytes) exhibited markedly higher expression of *GLUT3 (SLC2A3)* compared with other clusters. Although the precise mechanism responsible for the enhanced responsiveness of the EDV model to ascorbic acid remains unclear, the elevated expression of *SLC2A3 (GLUT3)* in HSE pericytes suggests a potential involvement of dehydroascorbic acid uptake and metabolism. Further investigation, such as single-cell-level analyses of intracellular ascorbic acid concentrations, will be required to elucidate this mechanism. We added these observation and discussion in Line 368-386 and 515-521.

3. The omission of immune cells which are key players in skin aging and homeostasis could increase physiological relevance of the model.

Our response:

Response to Reviewers

As rightly noted by Reviewer 2, immune cells play an essential role in skin aging and the maintenance of tissue homeostasis, highlighting the importance of incorporating them into future research models. In our EDV model, fibroblasts failed to clearly distinguish between the subpopulations of papillary secretory fibroblasts and pro-inflammatory fibroblasts, indicating that fibroblast heterogeneity was only partially recapitulated. The incorporation of immune cells could potentially facilitate the differentiation of pro-inflammatory fibroblasts. However, since establishing a reliable protocol for their integration will require substantial optimization, we plan to address this in future studies (Line 407-417).

4. The exclusive use of standard HUVECs may not fully capture the behavior of tissue-specific microvascular endothelial cells, potentially limiting the fidelity of the vascular niche.

In this study, we opted to use HUVECs as vascular endothelial cells due to their relative ease of expansion in culture. Consequently, we acknowledge the potential limitation in fully recapitulating the functions of tissue-specific endothelial cells. To address this concern, we have revised and expanded the Discussion section on lines 407–417.

Response to Reviewers

Reviewer #3

We have studied your comments carefully and found that you understood the value and significance of our study in this field. We are grateful for your evaluation and valuable suggestions for our manuscript. Our specific responses are listed below:

Major comments:

1. Are the key conclusions convincing?

The core claim-that tricellular interactions recapitulate dermal mesenchymal heterogeneity and enhance skin functionality-is well-supported by histology, immunohistochemistry, functional assays (TEWL, elasticity), and scRNA-seq.

2. Should the authors qualify some of their claims as preliminary or speculative, or remove them altogether?

The assertion that HSEs enable "identical" methodology to clinical studies (p. 2, line 29) is exaggerated. While elasticity was measured via Cutometer (used clinically), the model lacks immune/neural components and long-term stability for full translational equivalence.

Our Response:

Upon thorough consideration, we have deleted the statements that may be regarded as exaggerated (line 26-28 and 214-216).

3. Would additional experiments be essential to support the claims of the paper? Request additional experiments only where necessary for the paper as it is, and do not ask authors to open new lines of experimentation.

Adequacy of Experimental Evidence & Need for Additional Experiments: No essential control appears to be missing: the authors include conditions {plus minus}ascorbic acid and {plus minus}vascular cells to isolate those effects. One could suggest a few additional experiments to further bolster the conclusions, but they are not strictly required for the main message. For example, to pinpoint the contribution of each mesenchymal subset, the authors could engineer HSE variants lacking one component at a time (omit pericytes or use only papillary vs. only reticular fibroblasts) to see how each omission affects barrier or elasticity. This would directly confirm each cell type's role. However, such experiments may be technically involved (especially isolating pure papillary vs. reticular fibroblast populations and ensuring viability in 3D culture) and might be beyond the scope of a single study. Another possible extension could be mechanistic assays, such as examining specific molecular signals: e.g., testing if blocking known paracrine factors from pericytes or fibroblast subsets diminishes the observed improvements. Given that pericytes can secrete laminin-511 and other factors that promote keratinocyte growth, the authors might, in future work, explore whether such factors mediate the enhanced epidermal proliferation seen with the vascularized HSE. Overall, the current data are sufficiently convincing that additional experiments are not absolutely necessary for publication.

4. Are the suggested experiments realistic in terms of time and resources? It would help if you could add an estimated cost and time investment for substantial experiments-

Our response

We are deeply grateful for the reviewer's constructive feedback. As rightly pointed out, cell ablation and mechanistic assays utilizing signaling inhibitors to assess the contribution of individual mesenchymal subsets are indispensable for reinforcing our findings and claims. However, as the reviewer has also indicated, these experiments would require no less than four months to complete. Consequently,

Response to Reviewers

we have opted to forgo high-cost additional experiments such as the optimization of HSE construction protocols and inhibitor-based assays.

Alternatively, we performed pseudotime trajectory analysis using Monocle3 and CellChat analysis of the scRNA-seq dataset to gain further insights into the progenitor cells and molecular signals involved in pericyte differentiation. In summary, the pseudotime trajectory analysis indicated that Cluster #14 cells, which exhibited pericyte-like characteristics, originated from Cluster #9 fibroblasts. During this differentiation process, changes were observed in the expression of genes associated with angiogenesis, vascular maturation, and pericyte differentiation, including *KDR* (*VEGFR2*), *EGFL6*, *FGFR3*, and *TGM2* (Fig. 4d, and line 272-297). Future mechanistic studies targeting these genes may provide a more detailed understanding of the molecular mechanisms governing pericyte differentiation and their contribution to skin function.

6. Are the experiments adequately replicated and statistical analysis adequate?

The manuscript's data are presented in a manner that generally supports reproducibility. The authors state that all data are presented as "mean {plus minus} SD" (Methods, p.36). This is acceptable and clearly reported. However, I suggest that the authors consider using mean {plus minus} SEM for specific datasets where the primary goal is to assess statistical significance between groups - for example, for the Ki67-positive cell proliferation data (Fig. 6c) - as SEM better reflects the precision of the group mean for inferential comparisons. In contrast, for functional measures that inherently exhibit biological variation across samples (e.g., TEWL, skin elasticity), using mean {plus minus} SD remains fully appropriate, as SD reflects true inter-sample variability. To improve clarity and reproducibility, I encourage the authors to briefly state in the Methods or figure legends why SD or SEM is used in each case, in line with best practice guidelines.

Our Response:

We appreciate your guidance regarding the appropriate statistical analysis and data presentation. We reviewed all graphs to determine whether standard deviation (SD) or standard error of the mean (SEM) was the most suitable measure of variability, and we have added a brief explanation of the rationale for each choice in the corresponding figure legends.

Reviewer #3

Minor comments:

1. For Figure 4e, it would be helpful if the authors could clarify in the figure legend or Methods whether the heatmap shows log-normalized expression values (as derived from the Seurat object) or z-scored expression across cells or samples. This distinction affects the interpretation of relative versus absolute expression levels of the collagen and elastic fiber-related genes, which are central to the study's conclusions about ECM remodeling.

Our response:

Thank you for pointing out the inconsistency in data representation. We have revised the manuscript to clearly indicate that Fig. 4e presents the Z-score normalized average expression levels.

2. Typos: "factr" → "factor" (p. 16, line 244); "severl" → "several" (p. 22, line 367).

Our response

Thanks for pointing out the typo, we have corrected it.

Response to Reviewers

Response to Reviewer #4

We have studied your comments carefully and found that you understood the value and significance of our study in this field. We are grateful for your evaluation and valuable suggestions for our manuscript. Our specific responses are listed below:

Minor Points:

1. The human skin control in Fig. 1c seems thinner than normal and would suggest that the ED and EDV models are hyperproliferative. Replacing the control with one that shows normal thickness would prevent incorrect conclusions of the data.

Our response:

In accordance with the reviewer's suggestion, the display area of the human skin image in Fig. 1c has been modified.

2. KI67 and TEWL readings for human skin as controls for Fig. 2b-c would help gauge how the organoids perform and whether they are abnormal. What is the elasticity index for facial sagging?

Thank you for your valuable advice, which has deepened our understanding of the evaluation results of HSEs. We additionally quantified the number of Ki67-positive cells in human skin and incorporated these data into Fig. 2b. Epidermal basal cell proliferation in the EDV model was significantly enhanced compared with that in human skin. As it was not feasible to conduct additional human clinical measurements of transepidermal water loss (TEWL) during the revision period, we instead cited reference values for healthy human skin based on a systematic review and meta-analysis (Lines 178–182). Reported TEWL values were approximately 15.4 (13.9–17.0) g/m²/h for the cheek and 6.5 (6.2–6.8) g/m²/h for the forearm, which were comparable to those obtained from the ED and EDV models. Finally, a brief explanation of the measurement principle and the significance of the parameters used in the Cutometer-based skin elasticity analysis has been added to the manuscript (Lines 214–223).

3. Ascorbic acid utilizes SLC23A1 and SLC23A2 to transport across cell membranes. Are their expression more pronounced in cluster 14 fibroblasts? This would help connect the scRNA-seq data to the ascorbic acid experiments.

Our response:

We appreciate the valuable suggestions provided to investigate the mechanisms underlying the altered VC responsiveness observed in the EDV model.

We compared the gene expression levels of transporters potentially involved in regulating the intracellular ascorbic acid pool—SVCT1 (SLC23A1), SVCT2 (SLC23A2), GLUT1 (SLC2A1), GLUT3 (SLC2A3), GLUT4 (SLC2A4), and MRP4—across dermal mesenchymal cell clusters. Among these, SVCT2 (SLC23A2), GLUT1 (SLC2A1), and GLUT3 (SLC2A3) were detected, and GLUT3 (SLC2A3) expression was markedly higher in Cluster #14 fibroblasts than in other clusters. Given that GLUT3 facilitates cellular uptake of dehydroascorbic acid (DHA), which is subsequently reduced to ascorbic acid intracellularly, the elevated expression of GLUT3 may enhance the intracellular ascorbic acid pool and thereby potentiate the responsiveness of the EDV model to ascorbic acid. This interpretation has been incorporated into the revised manuscript (Lines 378-386 and 515-521.).

Response to Reviewers

4. There seems to be quite a bit of variability between replicant immunostains, in particular, vimentin in Fig. 3. Can the authors discuss this variability and whether any of the HSE organoid combinations reduced this variability?

Our response:

Thank you for your comments regarding the immunostaining. Upon careful re-examination of the staining results, we noted minor differences in staining intensity among samples processed on different experimental days; however, Vimentin staining in HSE fibroblasts was broadly comparable across all HSE conditions. In the CD36/Vim image set shown in Fig. 3a, the Vimentin channel had been displayed with a darker brightness/contrast range relative to the other panels. Therefore, we fine-tuned the display settings to ensure consistent presentation.

5. Please provide number of replicates throughout figure legends.

Our response:

Thank you for your valuable advice. We have added the number of replicates to all figure legends.

6. Line 148 states "E and EV models were transparent and extremely soft", should read "E and ED models".

Our response:

The photographic data for the EV and ED models in Fig. 1b was incorrect and has therefore been corrected. We sincerely apologize for our oversight. As it was actually the E and EV models that appeared transparent, the description in the text remains unchanged.

7. Line 150-151 states "In the E and EV models, an abnormal epidermis lacking a basal cell layer formed". The Krt5 staining in Figure 2 clearly shows a basal cell layer in these models, albeit abnormal. Stating that this the abnormal epidermis displayed a disrupted basal cell layer or columnar shape of basal cells were disrupted is more appropriate. In addition, these results do not show "crosstalk between NHEKs and NHDFs is essential for epithelialization" as the E and EV organoid models show epithelial stratification.

Our response:

We sincerely appreciate your insightful guidance regarding the accurate presentation of the histological analysis results. Accordingly, we have revised lines 166–168 in the Results section in line with your recommendations.

Dear Mr. Kimura,

Thank you for the submission of your revised manuscript to EMBO reports. Since my colleague Deniz Senyilmaz Tiebe is currently on maternity leave, I have taken over the handling of your manuscript. We have now received the full set of referee reports that is copied below. As you will see, all referees are very positive about the study and recommend publication.

From the editorial side, there are a few things that we need before we can proceed with the official acceptance of your study.

1) Please provide up to 5 keywords on the title page.

2) "Data and Code availability" should be renamed to Data Availability.

3) In the Data Availability section, please add URLs that resolve directly to the datasets deposited at SRA, BioStudies and Zenodo.

4) Please update the 'Conflict of interest' paragraph to our new 'Disclosure and competing interests statement'. For more information see <https://link.springer.com/journal/44319/submission-guidelines>

5) Regarding the Author Contributions, we now use CRediT to specify the contributions of each author in the journal submission system. CRediT replaces the author contribution section, which therefore needs to be removed from the manuscript text. You can use the free text box in our system if you wish to provide more detailed descriptions. See also guide to authors <https://link.springer.com/journal/44319/submission-guidelines>

6) Manuscript information on the title page needs to be removed from the manuscript.

7) You correctly labeled all preprint citations with [PREPRINT] in the reference list but please add the preprint label also to the in-text citations. e.g. (preprint: Ferreira et al, 2022)

8) Sole-Boldo et al, 2020 is cited twice in a row in line 665

9) McCarthy et al, 2017 is cited twice in a row in line 714

10) Are these data references, i.e., did you re-analyse datasets produced in these studies? If so, you would cite the primary paper, as you already did, and then add a so-called data citation with the same authors and year but a link to the repository and the published dataset. The reference in the reference list is marked with [DATASET] and the reference in the text with (data ref: ...).

11) Please remove the legends from the figures as they are already provided in the manuscript.

12) The nomenclature of EV figures is not correct in the figure files and legends. It needs to be Figure EV1, etc. instead of Expanded View Figure 1, etc.

13) Supplementary Fig. 2 is not a correct callout (line 324). I assume it should be updated to Figure EV2 since a callout for that one is missing. Please check.

14) Table EV1 is a dataset and needs to be updated to Dataset EV1 in all places (source file name, legend, title in the system, callout in the manuscript file). Sheet 2 in the file looks unconventional, having the legend and pseudotime trajectories on the same page. Could the legend be moved to a separate sheet?

15) Please remove this text from the Reagents and Tools table:

"Instructions: Please complete the relevant fields below, adding rows as needed. The following page provides an example of a completed table and additional instruction for entering your data in the table."

16) The text "Social Survey Research Information" in the Table has a grey background.

17) The manuscript sections should be in the following order: Title page - Abstract & Keywords - Introduction - Results - Discussion - Methods - Data Availability - Acknowledgments - Disclosure Statement & Competing Interests - References - Figure Legends - (Main Tables with legends if applicable) - Expanded View Figure Legends.

18) A part of the methods section from line 643 to line 660 reads very similar to the methods section on Data analysis in PMID 32327715.

("The integrated data were then used for standard cell clustering [...] Finally, for visualization we used the RunUMAP function with default parameters and 30 PCA dimensions") Please cite the related publication here if you used similar methods.

19) During our routine image checks, we noticed that the images across the figure set appear pixelated under analysis. This is a common result of converting original 16-bit TIFF images to RGB format for publication, and while not a cause for concern, it can sometimes give the impression of image alteration to critical readers.

To resolve this, please upload the figure set at a higher resolution.

20) Figure 1C: please indicate that the zoomed-in highlight images in the lower row are from the same images/samples shown in the upper row. Please add this explanation in the figure legend and best label the zoomed area in the overview image, e.g., with a dashed line. Please also add scale bars for both, the overview and the zoomed images.

21) Our data editors noted the following points in your figure legends, which I kindly ask you to address:

- Please provide the exact p values in the legends of figures 2b,c,g; 3b,d; 6c-e,h; EV-3b (unless these are < 0.0001).
- Please indicate the statistical test used for data analysis in the legends of figures 4e; EV-2d.
- Please define the box plots in terms of minima, maxima, centre, bounds of box and whiskers, and percentile in the legends of figures 3b,d; 6g; EV-3b.
- Please note that information related to n is missing in the legends of 6g.
- Please note that the scale bar needs to be defined for figure 2d.
- Please note that the black arrowheads are not defined in the legend of figure 1d. This needs to be rectified.

22) Finally, EMBO Reports papers are accompanied online by

A) a short (1-2 sentences) summary of the findings and their significance,

B) 2-3 bullet points highlighting key results and

C) a schematic summary figure that provides a sketch of the major findings (not a data image).

Please provide the summary figure as a separate file in PNG or JPG format at a size of 550x300-600 pixels (width x height).

Please note that the size is rather small and that text needs to be readable at the final size. Please send us this information along with the revised manuscript.

With kind regards,

=====

Referee #1:

The manuscript was sufficiently revised and all of my previous comments have been addressed by the authors. I have no further concerns and recommend the manuscript for publication.

Referee #2:

The authors have adequately addressed all the comments raised during the first revision. While the impact could have been greater if the mechanisms for inducing fibroblast heterogeneity in the vascularised models were directly tested, the study is technically sound and suitable for publication.

Referee #3:

The revised manuscript by Kimura et al. define how epidermal morphogenesis in human skin equivalents (HSE) differ by combining vascular endothelial cells, epidermal keratinocytes, and dermal fibroblasts using staining and single-cell RNA-sequencing (scRNA-seq). The authors have responded to critiques and have a much improved manuscript. I recommend acceptance.

Referee #1:

The manuscript was sufficiently revised and all of my previous comments have been addressed by the authors. I have no further concerns and recommend the manuscript for publication.

Referee #2:

The authors have adequately addressed all the comments raised during the first revision. While the impact could have been greater if the mechanisms for inducing fibroblast heterogeneity in the vascularised models were directly tested, the study is technically sound and suitable for publication.

Referee #3:

The revised manuscript by Kimura et al. define how epidermal morphogenesis in human skin equivalents (HSE) differ by combining vascular endothelial cells, epidermal keratinocytes, and dermal fibroblasts using staining and single-cell RNA-sequencing (scRNA-seq). The authors have responded to critiques and have a much improved manuscript. I recommend acceptance.

Rev_Com_number: RC-2025-03000

New_manu_number: EMBOR-2025-62369V2

Corr_author: Kimura

Title: The heterogeneity of dermal mesenchymal cells reproduced in skin equivalents regulates barrier function and elasticity

Point-by-point response

1) Please provide up to 5 keywords on the title page.

We added keywords in line 32.

2) "Data and Code availability" should be renamed to Data Availability.

Line 772 has been corrected.

3) In the Data Availability section, please add URLs that resolve directly to the datasets deposited at SRA, BioStudies and Zenodo.

The URLs has been added in line 774, 776, and 778)

4) Please update the 'Conflict of interest' paragraph to our new 'Disclosure and competing interests statement'. For more information see

<https://link.springer.com/journal/44319/submission-guidelines>

Line 786 has been corrected.

5) Regarding the Author Contributions, we now use CRediT to specify the contributions of each author in the journal submission system. CRediT replaces the author contribution section, which therefore needs to be removed from the manuscript text. You can use the free text box in our system if you wish to provide more detailed descriptions. See also guide to authors

<https://link.springer.com/journal/44319/submission-guidelines>

We deleted the author contribution section.

6) Manuscript information on the title page needs to be removed from the manuscript.

We deleted the manuscript information.

7) You correctly labeled all preprint citations with [PREPRINT] in the reference list but please add the preprint label also to the in-text citations. e.g. (preprint: Ferreira et al, 2022)

8) Sole-Boldo et al, 2020 is cited twice in a row in line 665

9) McCarthy et al, 2017 is cited twice in a row in line 714

Upon re-checking the reference list, we found that the '[PREPRINT]' notations were due to an error in our reference management software. We have corrected these entries in the revised manuscript.

10) Are these data references, i.e., did you re-analyse datasets produced in these studies? If so, you would cite the primary paper, as you already did, and then add a so-called data citation with the same authors and year but a link to the repository and the published dataset. The reference in the reference list is marked with [DATASET] and the reference in the text with (data ref:).

A portion of our data includes the re-analysis of publicly available datasets from the study by Boldo et al. As suggested, we have added the formal data citations to the manuscript to ensure proper attribution (Line 305-306, 662-664, 986-988, 1080-1083, and 1128-1131).

11) Please remove the legends from the figures as they are already provided in the manuscript.

We deleted the legends from figures.

12) The nomenclature of EV figures is not correct in the figure files and legends. It needs to be Figure EV1, etc. instead of Expanded View Figure 1, etc.

13) Supplementary Fig. 2 is not a correct callout (line 324). I assume it should be updated to Figure EV2 since a callout for that one is missing. Please check.

Line 322, 1122, 1127, 1142, and 1150 have been corrected.

14) Table EV1 is a dataset and needs to be updated to Dataset EV1 in all places (source file name, legend, title in the system, callout in the manuscript file). Sheet 2 in the file looks unconventional, having the legend and pseudotime trajectories on the same page. Could the legend be moved to a separate sheet?

Table EV1 has been substituted with Dataset EV1. We cleaned the Excel file by deleting irrelevant sheets and detaching the legend from the table. References to this data in the main text have also been revised (Line 289 and 1156).

15) Please remove this text from the Reagents and Tools table:

"Instructions: Please complete the relevant fields below, adding rows as needed. The following page provides an example of a completed table and additional instruction for entering your data in the table."

We deleted the text.

16) The text "Social Survey Research Information" in the Table has a grey background.

We collected it.

17) The manuscript sections should be in the following order: Title page - Abstract & Keywords - Introduction - Results - Discussion - Methods - Data Availability - Acknowledgments - Disclosure Statement & Competing Interests - References - Figure Legends - (Main Tables with legends if applicable) - Expanded View Figure Legends.

We collected it.

18) A part of the methods section from line 643 to line 660 reads very similar to the methods section on Data analysis in PMID 32327715.

("The integrated data were then used for standard cell clustering [...] Finally, for visualization we used the RunUMAP function with default parameters and 30 PCA dimensions") Please cite the related publication here if you used similar methods.

We added the references in line 647-653.

19) During our routine image checks, we noticed that the images across the figure set appear pixelated under analysis. This is a common result of converting original 16-bit TIFF images to RGB format for publication, and while not a cause for concern, it can sometimes give the impression of image alteration to critical readers.

To resolve this, please upload the figure set at a higher resolution.

We appreciate you pointing out the pixelation issues in our image data. We have replaced all figures with high-resolution versions and re-uploaded them to the system (<https://www.ebi.ac.uk/biostudies/studies/S-BSST2270>).

20) Figure 1C: please indicate that the zoomed-in highlight images in the lower row are from the

same images/samples shown in the upper row. Please add this explanation in the figure legend and best label the zoomed area in the overview image, e.g., with a dashed line. Please also add scale bars for both, the overview and the zoomed images.

In accordance with your advice, we have modified Fig. 1C and the corresponding legend (Lines 1022–1024).

21) Our data editors noted the following points in your figure legends, which I kindly ask you to address:

- Please provide the exact p values in the legends of figures 2b,c,g; 3b,d; 6c-e,h; EV-3b (unless these are < 0.0001).

- Please indicate the statistical test used for data analysis in the legends of figures 4e; EV-2d.

- Please define the box plots in terms of minima, maxima, centre, bounds of box and whiskers, and percentile in the legends of figures 3b,d; 6g; EV-3b.

- Please note that information related to n is missing in the legends of 6g.

- Please note that the scale bar needs to be defined for figure 2d.

- Please note that the black arrowheads are not defined in the legend of figure 1d. This needs to be rectified.

In accordance with your advice, we have modified all figures and the corresponding legend section.

22) Finally, EMBO Reports papers are accompanied online by

A) a short (1-2 sentences) summary of the findings and their significance.

B) 2-3 bullet points highlighting key results and

C) a schematic summary figure that provides a sketch of the major findings (not a data image).

Please provide the summary figure as a separate file in PNG or JPG format at a size of 550x300-600 pixels (width x height). Please note that the size is rather small and that text needs to be readable at the final size. Please send us this information along with the revised manuscript.

We attached a schematic summary figure and synopsis.

Dear Mr. Kimura

Thank you for the submission of your revised manuscript to our offices. I apologise for the delay in handling it, which was caused by a high number of submissions over the Christmas and New Year period causing a backlog that could not be cleared that fast.

We have meanwhile completed all checks and all seems fine. I however noticed that all quantifications in your study are based on technical replicates. The number of independent repeats is therefore 1 and the data are thus not adequately powered for statistical analysis. Please remove any statistical analysis from the figures and figure legends. In case you have independent repeats, please incorporate these in the analysis.

In addition, please remove the tags [PREPRINT] in the reference list.

Kind regards,

The authors have addressed the minor editorial requests.

Mr. Shun Kimura
Rohto Pharmaceutical Co., Ltd.
Basic Research and Development Division
Osaka
Japan

Dear Mr. Kimura,

Thank you for your patience while we have checked the final revision. I am very pleased to accept your manuscript for publication in the next available issue of EMBO reports. Thank you for your contribution to our journal.

You may qualify for financial assistance for your publication charges - either via a Springer Nature fully open access agreement or an EMBO initiative. Check your eligibility: <https://link.springer.com/journal/44319/how-to-publish-with-us>

Yours sincerely,

>>> Please note that it is EMBO Reports policy for the transcript of the editorial process (containing referee reports and your response letter) to be published as an online supplement to each paper. If you do NOT want this, you will need to inform the Editorial Office via email immediately. More information is available here: <https://link.springer.com/partners/embo-press/editorial-policies#Peer%20review>